



# Reviews and syntheses: Assessment of Biogeochemical Models in the Marine Environment

Kaltham A. Ismail, Maryam R. Al-Shehhi

Department of Civil, Infrastructure and Environmental Engineering, Khalifa University, Abu Dhabi, UAE

*Correspondence to*: Maryam R. Al-Shehhi (maryamr.alshehhi@ku.ac.ae)

**Abstract.** Marine biogeochemical models are key tools utilized to quantify numerous aspects of biogeochemistry including primary productivity, cycling of nutrients, redistribution of plankton, and variability of the carbon cycle in the ocean. These models are typically coupled to physical models with a horizontal resolution varying from few kilometers to more than 400 kilometers. Many of the existing biogeochemical models are commonly based on the NPZD model structure however, these models differ in their complexity determined by the number of state variables and the functional forms. Therefore, this review illustrates the types of the common biogeochemical models categorized based on the complexity levels and the governing equations. Then, applications of these models in several ecosystems of the world ocean are presented through a comprehensive assessment and evaluation of their performance in reproducing biogeochemical parameters such as chlorophyll-a, nutrients, as well as carbon and oxygen. In general, models based on functional group approach when coupled to high-resolution physical models show good estimates of surface nutrients such as nitrogen (N), phosphorous (P), silica (S) in global oceans with correlation coefficients (r) of $\geq 0.85$, $\geq 0.9$, and $\geq 0.78$ respectively. Similarly, NPZD based models coupled to suitable physical models are found to accurately reproduce N, P, and oxygen (O) with coefficients of determination ($R^2$) around 0.9 (for N & P) and $\sim > 0.9$ (for O) particularly in the Indian and Pacific waters. In addition, highest performance for iron prediction in global oceans is found with r values between 0.7 and 0.86 particularly by functional group approach models. However, chlorophyll-a prediction has shown varying performances by all types of models with r ranging from 0.55 and 0.9. So, applications of biogeochemical models are dependent on the features of the ecosystem and the purpose of the study. Therefore, the functional group approach models are mainly applied to investigate biogeochemical cycles while NPZD models are mainly used for physical-biological investigation.

## 1 Introduction

Modelling the biogeochemistry of the ocean is essential to improve our understanding of the primary productivity, eutrophication, and nutrients variability. The formal definition of biogeochemistry is to quantify the chemical species exchanged between earth system reservoirs along with transformations in these reservoirs. Thus, biogeochemistry focuses on carbon and nutrients cycling between the living and non-living compartments of the ocean (Dutkiewicz et al., 2020). This is translated into including the inorganic nutrients, detrital matter and the explicit representation of the living components such as phytoplankton and zooplankton in the biogeochemical modelling. In addition, the importance of the ocean circulation manifests in the redistribution of organic and inorganic pools hence representation of currents, temperature, mixing, salinity and density are also an integral part of the biogeochemical models and have a great impact on the primary productivity and nutrients distribution in the oceans (Heinze and Gehlen, 2013).



Therefore, the developed biogeochemical models are mainly based on the classical NPZD approach developed by
Fasham et al. (990) which stands for Nutrients, Phytoplankton, Zooplankton, and Detritus. These main four
compartments, can be categorized into biotic (e.g. phytoplankton, zooplankton, fishes, whales) and abiotic (e.g.
ammonium, nitrate, dissolved organic/inorganic carbon (DOC/DIC), particulate organic carbon (POC)) (Sarmiento et
al., 1993). As for biota, phytoplankton and zooplankton are the core parts of it where phytoplankton are autotrophic
organisms obtaining their energy from sunlight and can fix the carbon dioxide, and zooplankton are heterotrophic
organisms obtaining their energy source by consuming other organisms. For the abiotic components, in addition to
what was mentioned above, the biogeochemical models also consider the main limiting nutrient in the ocean which is
primarily the Dissolved Inorganic Nitrogen termed as DIN. The other important limiting elements that are also
considered include phosphate, iron and silicate (Lachkar et al., 2020). The representation of these compartments is
governed by one or more state variables which can be used to define the trophic levels of the pelagic ecosystems'
evolution (Heinze and Gehlen, 2013).
Several substantial biochemical parameters have been studied in various ecosystems of the global oceans using
different types of biogeochemical models, these parameters include chlorophyll-a, macronutrients (nitrate, phosphate,
silicate), micronutrients (Fe), carbon and oxygen cycles. Chlorophyll-a is typically used as a metric of biomass
concentration instead of carbon biomass in the ocean due to its unique optical properties and it is one of the widely
studied parameter in the biogeochemical modelling. The level of this parameter is affected by several basic factors
including: the solar radiation intensity penetrating the water column, dissolved nutrients gradients with depth,
temperature, and the mixed layer depth (Sverdrup, 1953; Wroblewski et al., 1988). Although the chlorophyll-a to
carbon and nutrient ratio (Chl: C:nutrient) is highly variable due to an acclimatize response to changes in
environmental conditions such as irradiance, temperature, and nutrient availability, this flexibility is neglected by
many large-scale biogeochemical models for the sake of simplicity and lowering complexity (Anugerahanti et al.,
2021). Whereas Macronutrients such as Nitrate ($NO_3$), Silicate ($SiO_3$), and Phosphate ($PO_4$) play a critical role in
phytoplankton growth and ocean dynamics and these nutrients are considered to be key limiting nutrients impacting
oceanic primary productivity; however, iron is recently well established to be also one of the key limiting nutrients
highly impacting phytoplankton dynamics and primary productivity. These limiting nutrients can be supplied to the
ocean through several sources including: dust deposition from atmosphere, riverine inputs, sea ice, sediment
mobilization, as well as hydrothermal vents (Aumont et al., 2015). Unlike other nutrients, iron sources in the ocean



65 mainly come from the atmosphere, transported as aerosols and commonly related to soil dust. Different phytoplankton

66 groups have different sensitivity to iron limitation, for example diatoms exhibited a large sensitivity to iron limitation

67 (Gregg et al., 2003) compared to other phytoplankton types. Likewise, carbon is the primary element in the

68 photosynthesis process carried out by autotrophs mainly phytoplankton in the surface of the ocean. It is also the energy

69 source for many aerobic heterotrophs and autotrophs living in the ocean. The inorganic form of carbon can be oxidized

70 through remineralization to form inorganic sources to be utilised by photo synthesizers. While the latter convert the

71 inorganic form back into organics for the heterotrophs. So, the atmospheric carbon dioxide is regulated by biological

72 carbon pump which is highly impacted by the role of zooplankton in the ocean (Cavan et al., 2017). Oxygen is a by-

73 product of photosynthesis and can be dissolved into the ocean from the atmosphere. This parameter is important for

74 aerobic heterotrophs living in the ocean and its reduction in the ocean can lead to amplify denitrification creating

75 oxygen minimum zone (OMZ) which is found in some regions of the global oceans (Lachkar et al., 2016, 2019, 2020).

76 These parameters have been modelled by several bio-geochemical models to better understand the ocean ecosystems

77 and therefore the aim of this work is to describe the most common biogeochemical models and carry out a complete

78 assessment of these biogeochemical models in estimating the aforementioned biochemical properties in different

79 ecosystems. This includes reporting the performance of the models, strengths, uncertainties and limitations. This

80 review begins with the models' section describing their components, assumptions and structure as well as examples

81 of well-known models developed based on these approaches. Then, the major modelled parameters studied in the

82 several ecosystems of the global oceans are discussed.

83 **2 Biogeochemical Modelling Approaches**
84

85 The existing biogeochemical models are categorized here into three types in terms of complexity, the number of state

86 variables, and the governing equations that is formed based on the functional forms, as follow,

87 **2.1 Classical NPZD approach**
88

89 This approach basically considers a single variable for each compartment (nutrients - phytoplankton - zooplankton –

90 detritus sometimes includes bacteria) neglecting the differences between the species (Evans et al., 1985; Fasham et

91 al., 1990, 1993; Franks P, 2002). In this approach, nitrogen is typically considered a limiting nutrient and detritus

92 component account for the organic matter pool which are derived from fecal materials and / non-assimilated fraction





of grazing by zooplankton and phytoplankton decay. This detritus is recycled through two ways which are utilization
by bacteria and degradation of dissolved organic nitrogen/zooplankton assimilation (Leles et al., 2016).
The general form of NPZD is presented in equations 1-4.
$\frac{dN}{dt} = -f(I)g(N)P + R(D)D$                                                                     (1)
$\frac{dP}{dt} = f(I)g(N)P - h(P)Z - i(P)P$                                                              (2)
$\frac{dZ}{dt} = \gamma\, Z\, h(P) - j(Z)Z$                                                              (3)
$\frac{dD}{dt} = i(P)P + j(Z)Z + (1-\gamma)h(P)Z - R(D)D$                                                (4)
Five transfer equations are involved in the model including: light limitation (phytoplankton response to
light/irradiance) $f(I)$; nutrient limitation (uptake of nutrients by phytoplankton) $g(N)$; grazing by zooplankton $h(P)$;
loss terms due to excretion, death, and predation by other organisms $i(P)P, j(Z)Z$; degradation of detritus $R(D)$.
The zooplankton assimilation is termed as $\gamma$ which is commonly modelled by a simple linear function of food ingestion
(Franks P, 2002). The functional forms representation of phytoplankton response to incident light ranges from a simple
linear form to nonlinear functions including saturation and photo-inhibition response (see Table 1). The Michaelis-
Menten/Monod saturation function is the most applied form of nutrient uptake by phytoplankton which can relate the
growth rates to the concentration of a limiting nutrient (Dugdale, 1967). The dependency of the growth on nutrient
concentration is regulated by two kinetic parameters which represent the population traits: the maximum utilization
rate, $V_{max}$ ; and the affinity constant, $k$, which presents an organism ability to capture nutrient ions at low nutrient
concentration $N$. The phytoplankton acclimation determines the ability of the cell to adapt its kinetic parameters in
response to changes in environmental conditions. So, if $V_{max}$ is constant then the acclimation will be discarded in the
Michaelis-Menten formulation because the maximum uptake rate is associated with the total number of uptake sites
of the cell (Bonachela et al., 2015). However, the Michaelis-Menten assumption was argued by (Droop, 1973, 1983)
which has assumed that the growth rate is more likely dependent on the internal content of the nutrients than the
external concentration showing luxury uptake of nutrients (utilization of non-limiting nutrient exceeding the level
required for growth) (Cherif and Loreau, 2010). Hence, the growth of phytoplankton is described by a function of
internal concentration (Quota model), as shown in Table 2.  In addition, it is argued that the growth rate is determined



by the most limiting process either photosynthesis or nutrient uptake permitting  for switching between the two limiting
processes based on the conditions (Franks P, 2002). Whereas zooplankton functional response is typically modelled
with a simple functional form represented by (HOLLING CS, 1959): Holling Type I with linear function, Holling
Type II with hyperbolic curve like Monod function accounting for saturation; and Holling Type III accounting for
saturation and switching when the prey is low in density (sigmoidal). The zooplankton is considered as the closure
term in plankton models and the zooplankton grazing functional forms impact model outputs greatly. For example,
high oscillations of the states over time (destabilization effect) are determined using type II while steady state
(stability) is easily obtained with type III and no impact on model stability was determined with type I. Phytoplankton
and zooplankton mortality functions ranges from linear to non-linear forms (Tables 3 and 4).
Modifications have been also made in the NPZD approaches such as by replacing the bacteria compartment with
chlorophyll-a to enhance the estimation of nitrogen flux (Fennel et al., 2006)    and introducing a nitrogen based
nutrient-phytoplankton-heterotroph model which is of intermediate complexity with respect to Fasham &
McGillicuddy models (Fasham et al., 1990; McGillicuddy et al., 1995). The number of compartments has also been
increased including more variables (plankton species as well as nutrients) as seen in Chai and Leonard models which
are based on five and nine compartments NPZD models respectively (Chai et al., 1996; Leonard et al., 1999).
Nevertheless, Galbraith et al. (2009) has developed a model based on NPZD but with Light Iron Nutrients and Gasses
called BLING model, This model can isolate the global impact of iron on maximum light-saturated photosynthesis
rates from photosynthetic efficiency. It considers an implicit representation of phytoplankton which is determined
from the growth rate of phytoplankton. The iron representation doesn't rely on Liebig law of the minimum that is
typical in the biogeochemical models, however, the nutrient-light co-limitation is incorporated in accordance with
field and laboratory measurements of phytoplankton. There have been other extensions of the classical NPZD which
have been applied regionally in (Doney et al., 1996; Fennel et al., 2001; Hinckley et al., 2009; Hood et al., 2003;
Kearney et al., 2020; McCreary et al., 1996; McCreary et al., 2001).
**2.2 Carbon cycle-based approach**

In this approach, the marine biota model is introduced into a full ocean carbon cycle model to study the impact of
biology on the oceanic carbon cycle. The carbon cycle model typically includes dissolved inorganic carbon and total
alkalinity components. An example of this approach is the Hamburg model of the oceanic carbon cycle (HAMOCC)





developed by (Maier-Reimer and Hasselmann, 1987) which is a pure inorganic carbon cycle model and was utilised
to evaluate both the 12C cycle and the ocean model residence time properties. The model neglects biological sources
and sinks. Therefore, it has been used as a reference for numerical experiments interpretation with extensions
performed by (Bacastow and Maier-Reimer, 1990; Heinze and Maier-Reimer, 1991; Maier-Reimer, 1993) to include
the marine biota and ecosystem processes. Bacastow & Maier-Reimer (1990) has included the first order
representation of the ocean plankton impacts on the ocean global inorganic oceanic carbon cycle model. While the
first ocean carbon cycle model featuring the representation of marine ecosystem explicitly was given by (Six and
Maier-Reimer, 1996). This latter is based on an extended NPZD model which includes five compartments: single
phytoplankton, single zooplankton, detritus, dissolved organic carbon (DOC), and single nutrient (phosphate).
Equations 5-11 represent the rate of change of nutrients as an example of the carbon components including: DOC and
particulate organic carbon (POC) embedded in the plankton model as described by (Six and Maier-Reimer, 1996).
$$R_{C:P} \frac{dN}{dt} = -phytoplankton\ growth$$
$$+remineralisation\ from\ herbivores$$
$$+remineralisation\ from\ carnivores$$
$$+DOC\ degradation + POC\ remineralisation$$
(5)

where $R_{C:P}$ represents the Redfield ratio of carbon to phosphate. Whereas phytoplankton and zooplankton are
described as follows:
$$\frac{dP}{dt} = phytoplankton\ growth$$
$$-loss\ due\ to\ grazing - natural\ decay$$
$$-exudation\ of\ DOC$$
(6)

$$\frac{dZ}{dt} = zooplankton\ growth$$
$$-loss\ due\ to\ grazing\ by\ carnivores - DOC\ excretion$$
(7)

Then, the carbon components (DOC & POC) are modelled as follows:
$$\frac{dDOC}{dt} = \gamma_P (P - P_{min}) + \gamma_Z (Z - Z_{min}) - r_{doc}(N)DOC$$
(8)

where the first term represents the DOC exudation from phytoplankton; the second term represents the DOC excretion
from zooplankton; and the last term represents DOC degradation.





$$\frac{\mathrm{d}POC}{\mathrm{d}t} = F(X) - l(O_2)$$     (9)
where $l(O_2)$ represents remineralization of POC (for X = $(d_p, d_z, \epsilon_{her}, \epsilon_{can}, P, Z, z)$ and F(X) is the flux of dead organic
carbon to the ocean interior
$F(X) = 0$ ; for $0 < z < 100\text{m}$
Otherwise
$$F(X) = \text{TPP} \frac{\partial}{\partial z} \left(\frac{z}{100m}\right)^{-0.8}$$     (10)
Where TPP is the total particle production including the particles from natural decay as well as fecal pellet production
in the euphotic zone:

$$\text{TPP} = \int_0^{100\text{m}} \big((1 - \text{ zinges })(1 - \epsilon_{her}) \; growth \; of \; zooplankton$$
$$+ phytoplankton \; mortality$$
$$+ (1 - \epsilon_{can}) \; zooplankton \; mortality \big)$$     (11)

| Parameter | Symbol |
|---|---|
| Mortality rate of phytoplankton | $d_p$ |
| Mortality rate of zooplankton | $d_z$ |
| Ingestion as fecal pellets from herbivores | $\epsilon_{her}$ |
| Ingestion as fecal pellets from carnivores | $\epsilon_{can}$ |
| Assimilation efficiency | zinges |
| Phytoplankton | $P$ |
| Zooplankton | $Z$ |
| Nutrient | $N$ |
| Depth | $z$ |


The Hadley Centre Ocean Carbon Cycle (HadOCC) model is another example of Carbon cycle-based approach that
is initially developed for global ocean carbon cycle modelling studies (Cox et al., 2000). The model simulates the
important aspects of carbonate chemistry, the export and production of biology. Several tracers are included to model
the carbon cycle including dissolved inorganic carbon, total alkalinity, single nutrient (nitrogen), oxygen, single
phytoplankton, single zooplankton, as well as detritus (Palmer and Totterdell, 2001).





**182**     **2.3 Phytoplankton Functional group approach (PFT)**

**183**

**184**   This approach includes different plankton functional types (PFTs) making it the most intricate model with at least 15

**185**   state variables relative to the other model approaches (Gregg, 2000; Gregg et al., 2003; Moore et al., 2004; Le Quéré

**186**   et al., 2005). The major plankton functional types include mesozooplankton, protozooplankton, diatoms

**187**   (phytoplankton silicifiers), phaeocystis, nitrogen fixers, coccolithophores, picoheterotrophs and each of these groups

**188**   function differently in terms of their roles in biogeochemical cycles (Hood et al., 2006). These functional traits that

**189**   reflect the functions and biochemical pathways are defined by how the cell uses energy and nutrients. The classical

**190**   NPZD doesn't consider these functional types in which the aggregation of taxonomic and functional organisms in

**191**   ocean ecosystems is only considered. Therefore, in the PFTs based approach, species are grouped based on their

**192**   common ecological and biogeochemical functions (Hood et al., 2006; Le Quéré et al., 2005). Equations 12-14 present

**193**   the general form of this approach where several phytoplankton types $P_j$ are nourished by various nutrients $N_i$ and

**194**   grazed by many zooplankton types $Z_{ki}$ as follow,

**195**   $$\frac{dN_i}{dt} = -\sum_j \left[ \mu_j P_j M_{ij} \right] + S_{N_i} \tag{12}$$

**196**   $$\frac{dP_j}{dt} = \mu_j P_j - m_j^P P_j - \sum_k \left[ g_{jk} Z_{k,i=1} \right]$$
$$- \frac{\partial \left( w_j^P P_j \right)}{\partial z} \tag{13}$$

**197**   $$\frac{dZ_{ki}}{dt} = Z_{ki} \sum_j \left[ \zeta_{jk} g_{jk} M_{ij} \right] - m_k^Z Z_{ki} \tag{14}$$

| Parameter | Symbol |
|---|---|
| Growth rate of phytoplankton j | $\mu_j$ |
| Matrix of the ratio of element i to currency (which can be phosphorous, nitrogen, etc.) | $M_{ij}$ |
| Sources of tracer $N_i$ | $S_{N_i}$ |
| Rate of mortality/excretion of phytoplankton j | $m_j^P$ |
| Grazing of zooplankton k on phytoplankton j | $g_{jk}$ |
| Sinking rate for phytoplankton j | $w_j^P$ |
| Grazing efficiency of zooplankton k on phytoplankton j | $\zeta_{jk}$ |
| Rate of mortality/excretion of zooplankton k | $m_k^Z$ |

**198**



The common PFTs' models include the European Regional Seas Ecosystem Model versions (1 & 2) : ERSEM I,
ERSEM II (Baretta-Bekker et al., 1997; Baretta et al., 1995; Blackford et al., 2004) which are based on a generic lower
trophic approach developed to study the cycling of carbon as well as nutrients. In ERSEM, the ecosystem is divided
into three functional types in which the biotic groups are classified by their functional role not by species. For instance,
phytoplankton as producers; bacteria as decomposers; zooplankton as consumers which are further subdivided based
on trait-size and uptake of silica to represent a food web. The functional group dynamics are represented by including
population processes such as growth, migration, and mortality as well as physiological processes such as ingestion,
respiration, excretion, and egestion. The phytoplankton groups involve pico-phytoplankton, nano-phytoplankton,
diatoms, and non-siliceous macro-phytoplankton, while zooplankton groups include micro-zooplankton, heterotrophic
nano-flagellates, and meso-zooplankton. The ERSEM model was initially applied to the North Sea to study the
seasonal cycling of nutrients (N, P, S, C). A further modification has been made to the ERSEM to produce another
version called Biogeochemical Flux Model (BFM). This latter accounts for the Chemical Functional Families (CFFs)
in the state variables. The CFFs is split into living, non-living, and inorganic states (Vichi et al., 2007).
The Pelagic Interactions Scheme for Carbon and Ecosystem Studies (PISCES) model is another example of PFTs
based model that is a modified version of HAMOCC considering 24 state variables including $NO_3$, $NH_4$, $PO_4$, $SiO_2$,
Fe; small phytoplankton, large phytoplankton, small zooplankton, large zooplankton, DOM, small detritus, and large
detritus (O. Aumont et al., 2003). PISCES model has been extensively used to study several ecosystems and widely
applied in more than hundred studies that are based on this approach either directly or indirectly (Aumont et al., 2015).
Likewise, the NASA Ocean Biogeochemical Model (NOBM) is another type of PFTs based model originally coupled
to the Ocean-Atmosphere Spectral Irradiance Model (OASIM) (Gregg, 2001; Gregg et al., 2009). NOBM comprises
of four phytoplankton groups, four nutrient groups (nitrate, regenerated ammonium, silica, and iron), a single
zooplankton group, and three detrital pools (organic material storage, sinking, and remineralization) (Gregg, 2000;
Das et al., 2019; Gregg et al., 2003; Gregg and Casey, 2007; Trull et al., 2018) (Gregg, 2001) (Gregg et al., 2003).
Additionally, the PlankTOM biogeochemical model is a dynamic ocean model describing lower trophic level of
marine ecosystems. This model has several extensions through varying in the number of PFTs resolved. For example,
six PFTs: diatoms, coccolithophores, mixed phytoplankton, bacteria, protozooplankton and meso-zooplankton are
included in PlankTOM6 (Le Quéré et al., 2005). However, additional PFTs such as nitrogen fixers, Phaeocystis,
picophytoplankton and macro-zooplankton are added into PlankTOM10 (Buitenhuis et al., 2013) to evaluate the



interactions between climate and ocean biogeochemistry with the wide use of data synthesis for parametrizations of
the PFTs growth rates (Kwiatkowski et al., 2014) .PlankTOM resolves the cycle of carbon (C), nitrogen (N), oxygen
(O), phosphorous (P), Silicon (Si), iron (Fe) cycle, three types of organic detritus, air sea fluxes of CO2, O2, Dimethyl
sulphide (DMS) and N2O.
Moreover, the Model of Ecosystem Dynamics, nutrient Utilization, Sequestration and Acidification (MEDUSA) is
developed by (Yool et al., 2011, 2013) is a model of intermediate complexity, constructed beyond the standard NPZD
formulations. The biogeochemical cycles of iron, silicon, and nitrogen as well as small and large plankton size classes
are included in this model. In this specific model, an explicit representation of internal chlorophyll quotas is included
to allow for light acclimation. The key focus of MEDUSA is the biological sequestration of carbon in the deep ocean.
The model is developed to study the biogeochemical response particularly of the so-called biological pump to human-
induced driven change in the global ocean. Nevertheless, the tracers of phytoplankton with allometric zooplankton
(TOPAZ) model is based on the interactions between the biogeochemical and the carbon cycles including two
dissolved organic matter forms, dissolved inorganic species for coupled carbon (C), nitorgen (N), Phosphorous (P),
Silica (S), Iron (Fe), calcium carbonate (CaCO3), dissolved oxygen (O2) heterotroph, lithogenic cycling. Additionally,
processes such as gas exchange, scavenging, atmospheric deposition, denitrification and nitrogen fixation, sediment
processes, and river inputs were included (Dunne et al., 2010). This model has been implemented in several studies
such as (Bronselaer et al., 2020; Jung, et al., 2019; Sharada et al., 2020). The extended version of TOPAZ is the
Carbon, Ocean Biogeochemistry and Lower Trophics (COBALT) which was developed to improve the planktonic
food web dynamics resolution to examine the influence of climate on the flow of energy from phytoplankton to fish
(Stock et al., 2014). The planktonic food web representation in TOPAZ is highly idealized including an implicit
representation of zooplankton and bacteria hence, an implicit modelling of the impacts of these groups on
phytoplankton and biogeochemical processes were applied. Therefore, these limitations were addressed in COBALT
by including three explicit zooplankton groups, bacteria with a mechanistic parametrizations of the impacts of these
groups on biogeochemistry (Stock et al., 2014b). COBALT has resolved the global scale cycles of nitrogen, carbon,
phosphate, silicate, iron, calcium carbonate, oxygen and lithogenic material where small and large phytoplankton are
involved. Nevertheless, DARWIN biogeochemical model is a more complex PFTs based model consisting of 78
phytoplankton types, heterotrophs, organic and inorganic forms of nitrogen, phosphorous, iron, and silica. This model
was developed first to study the phytoplankton distribution especially for the cyanobacterium Prochlorococcus species



by (Follows et al., 2007). The model was coupled with the  general circulation model in (Wunsch and Heimbach,
2007) and was initially applied for the global distributions of phytoplankton and physiological traits. It has been
applied in a followed study which has considered more biogeochemical components and enhancement of optical
properties (Dutkiewicz et al., 2015; Lo et al., 2019). Nonetheless, Regulated ecosystem model (REcoM) based on
functional group approach (two phytoplankton group: diatoms and nanophytoplankton; one class of zooplankton) is
based on the Quota model in which the internal phytoplankton cells stoichiometry is affected by the conditions of
temperature, light, and nutrients (Schourup-Kristensen et al., 2014). REcoM has also been commonly used in the
Southern Ocean studies (Hauck and Völker, 2015; Losch et al., 2014; Taylor et al., 2013).
The complexity of the aforementioned PFTs models depends on number of the independent elements along with the
number of PFTs considered. As regards the PFTs, simple models include one PFT which is of single phytoplankton
and single zooplankton such as in HadOCC (Palmer and Totterdell, 2001). Simple models can also include two to
three PFTs such as MEDUSA (Yool et al., 2013) and PISCES (Aumont et al., 2015). However, as the number of PFTs
increases, the complexity of the model increases as well. As for the average elemental composition of particulate
matter, it is constrained in the sea despite of the variations in the carbon to chlorophyll (C/Chl) ratios (Anugerahanti
et al., 2021). The commonly used average proportion of the main elements in phytoplankton is: 106 C (carbon): 16 N
(nitrogen): 1 P (phosphorous) (by atoms) and these proportionalities are termed as Redfield ratios (Redfield, 1933).
Generally, adding complexity to the model doesn't necessarily improve the model skill, as has been proven in several
studies which compare models with different complexities (Friedrichs et al., 2007; Kriest et al., 2010; Kwiatkowski
et al., 2014; Ward et al., 2013; Xiao and Friedrichs, 2014).
**3 Determination of the biochemical parameters**

The aforementioned models have been applied to resolve the biochemical properties including Chlorophyll-a,
Macronutrients (N,P,S), Micronutrients (Fe), Carbon and Oxygen in different ecosystems. Detailed assessments of
the capabilities of these models are provided here and summarized in Table 5,
**3.1 Chlorophyll-a**

Chlorophyll-a concentrations have been determined using the models described above, however, the PFTs based
models are found to offer more accurate estimates of Chlorophyll-a concentrations by distinguishing the



phytoplankton types. The PFT based model PlankTOM, for instance, was used to evaluate the role of grazing versus
iron limitation in the low chlorophyll content (HNLC) areas of the Southern Ocean (Le Quéré et al., 2016). The
PlankTOM was able to produce reasonable surface chlorophyll-a estimates with correlation coefficient (r) around 0.8
especially in the summer season when the macro-zooplankton grazing was explicitly involved. PlankTOM5.3 has
shown large improvements of the interannual variation of surface chlorophyll-a relative to PlankTOM5.2 in the global
oceans (with residual sum of squares RSS = -13%) (Buitenhuis et al., 2013) by including a new photosynthesis
formulation with a representation of iron-light colimitation (Geider et al., 1998) in their fixed stoichiometry model.
PlankTOM10 was also compared to PlankTOM6 and applied in the Southern Ocean (Le Quéré et al., 2016). This new
version has similar formulations to the previous versions of the model except that it included more phytoplankton
groups. Both models exhibit similar results for the surface chlorophyll-a concentrations (r ~ 0.8), primary and export
production except that PlankTOM6 was unable to reproduce the observed low chlorophyll-a contents in summer
season in the Southern Ocean due to slightly deeper mixed-layer depth in the summertime. Overall, PlankTOM10 has
shown slightly better performance than PlankTOM6 in terms of surface chlorophyll-a distribution (bias% = 1.2%),
whereas the distribution of surface nutrients has been slightly lower by 5% and 2.5% for nitrogen and silica (except
for phosphate which shows similar performance r ~0.9). Other PFTs based models including ERSEM, DARWIN,
TOPAZ, PISCES, BLING and NOBM have been applied to study chlorophyll-a distribution in the surface and deep
oceans. ERSEM has been applied to study chlorophyll-a  dynamics in the Mediterranean and has shown good r value
of 0.64 for the spatial distribution of the simulated and observed chlorophyll-a. However, a relatively larger bias with
root mean square difference (RMSD) of 0.78 was obtained for the annual mean spatial variability due to the absence
of cyclonic gyres of the Rhodes and South Adriatic causing intermittent blooms. In addition, the PFTs based model
DARWIN coupled with MITgcm in (Dutkiewicz et al., 2015) apprehended large spatial variability in chlorophyll-a
for the global oceans with low r  around  0.55 and have shown overestimation in particularly the Southern Ocean and
higher latitudes. DARWIN was customized to study phytoplankton distribution  in the Southern Ocean (Lo et al.,
2019) which has included the abundance of coccolithophores which was improved through increasing the affinity for
nutrients as well as coccolithophores grazing control. Two distinct size classes of diatoms (small & large) were added,
and two different life stages were considered for Phaeocystis (single cell vs colonial). The improvements have
increased the agreement between the simulated coccolithophores and diatoms with the in-situ data. However, the
model inaccurately has simulated diatoms and haptophytes in the Ross Sea and has overestimated the small non-





silicified phytoplankton with general mean absolute error for diatoms and haptophytes are 0.74 mg m$^{-3}$ and 0.22 mg
m$^{-3}$ respectively. This inconsistency can be attributed to inaccuracy in representing PFT phenology and distribution.
Representation of co-existence within coccolithophores and Phaeocystis remains a challenge and any small changes
in DARWIN physiological parameters led to either Phaeocystis or coccolithophores loss. In addition, the sea-ice algae
representation has not been explicitly represented which may not work well in region where ice exists.
Concentrated Chlorophyll-a condition (i.e., Phytoplankton blooms) was also captured in the middle latitudes of the
Northern and Southern Hemispheres as well as in the tropical Pacific by both models (NEMO-TOPAZ and NEMO-
PISCES) indicating El Niño-Southern Oscillation (ENSO) condition. Both models showed an overall r between 0.6-
0.9 across all oceans. While the zonal averaged Chlorophyll-a was overpredicted (by ~67%) from 30°N to 45°N in
both models especially in the Pacific Ocean east of Japan which is due to mainly an error in the Kuroshio Current path
seen in low resolution models (Jung et al., 2019a). The physical model NEMO has simulated a relatively thicker mixed
layer which in turn simulated bigger spring blooms in this area creating a positive bias in Chlorophyll-a values in the
mid-latitudes of the Northern Hemispheres in biogeochemical models.
In contrast, underestimation of surface Chlorophyll-a in the equatorial Atlantic Ocean (bias ~-0.2 μg kg$^{-1}$) and the
Arabian Sea (bias ~-0.4 μg kg$^{-1}$) was also found in both models and these two areas encountering mesoscale and sub-
mesoscale processes impacting the biogeochemistry. NEMO-PISCES with the use of higher resolution grid than the
one used in (Jung et al., 2019a) presented better Chlorophyll-a distribution in the Arabian Sea  with an average value
of ~1.3 mg Chl m$^{-3}$ d (Koné et al., 2009). Although both models overpredicted the surface chlorophyll-a in the
Southern and topical Pacific (STD ~0.26 μg kg$^{-1}$), the NEMO-TOPAZ showed a larger bias over the equator (STD ~
0.22 μg kg$^{-1}$) for the surface Chlorophyll-a than PISCES-NEMO which is proved to be caused by the high atmospheric
iron deposition in TOPAZ, which is then replaced with PISCES data resulting in lowering the bias for surface
Chlorophyll-a. Hence, sensitivity experiments on atmospheric iron deposition can be a good task to improve the
surface chlorophyll-a distributions in simulations.
Similarly, the PFTs based model NOBM was able to predict surface Chlorophyll-a level (r > 0.7) in global oceans
(Gregg et al., 2003). However, the correct species abundance was not well identified by the model due to disparities
between the model and observations in particular the Indian Ocean where the observations were mainly concentrated
in the Arabian Sea (the model is modestly dominated by diatoms whereas observations are dominated by
cyanobacteria). This might have caused by the strong upwelling in the model thus increasing nutrients concentrations
which trigger faster growing organisms such as diatoms.
Nevertheless, BLING-NEMO coupled model was used to study the high Chlorophyll-a levels (i.e. blooms)
(Castro de la Guardia et al., 2019) in which the spring bloom in the Barents Sea (BS) was underestimated by 1.7 mg
chl m-3 while the autumn bloom underpredicted by 0.7 mg chl m$^{-3}$. This deviation might be attributed to the lack of
nutrients from riverine input by the BLING-NEMO coupled model. Whereas the concentration representing the spring
and autumn bloom in the Labrador Sea (LS) has shown an agreement with the observed seasonality, which is
comparable to that of the satellite data. However, the Chlorophyll-a content is slightly overestimated by 0.2 mg chl
m$^{-3}$ during February-April due to an earlier start of the spring bloom in the simulation. Furthermore, the BLING-
NEMO model has mistakenly predicted the spring blooms in March instead of April in the Hudson Bay (HB) and
Baffin Bay (BB). In these two bays, the spring bloom was slightly overpredicted by 0.5 mg chl m$^{-3}$ and ~ 0.3 mg chl
m$^{-3}$ in the BB and HB respectively. This discrepancy might be attributed to the underprediction of sea ice
concentration.
Chlorophyll-a has been also derived by the carbon cycle and simple NPZD models. For example, the carbon cycle
based model: HAMOCC5 was able to reproduce Chlorophyll-a with a value of 0.05 mg Ch m-3 in the oligotrophic
subtropical gyres (Aumont et al., 2003) with a bias of 0.24 mg chl m$^{-3}$. Unlike the earlier version of the model
(HAMOCC3.1) which has shown a higher concentrations of Chlorophyll-a compared to observation in these regions
(Six and Maier-Reimer, 1996). This version of HAMOCC5 is an extension of HAMOCC3.1 (Six and Maier-Reimer,
1996) with the inclusion of iron and silicate limitation along with the phosphate. HAMOCC5 with its coarse resolution
cannot resolve the coastal upwellings in productive regions such as Peru upwelling. The improvement of the
HAMOCC was mainly in making the Chl: C ratio variable which is decreasing in the centre of the subtropical gyres
to values about 1:150 while in HAMOCC3.1 this ratio remained constant at 1:60. Therefore, this model with this
improvement as well as iron and silicate limiting nutrients inclusion improved the representation of chlorophyll-a
content in subtropical gyres (around 0.2-0.25 mg m$^{-3}$).
As for the simple NPZD models, they have shown a better representation of chlorophyll-a when coupled to a high-
resolution physical model as well as including correct physics representations. For example, the involvement of tides
in ROMS in (Fennel et al., 2008) has improved the chlorophyll-a representations over Georges Bank which was not



presented in the previous model (Fennel et al., 2006). The same model of (Fennel et al., 2008) was improved by adding

dissolved organic matter (DOM) module along with the other model components (Druon et al., 2010) to study the

DOM dynamics in the Northeastern U.S. continental shelves which showed a well agreement of high chlorophyll-a

concentrations with the satellite data particularly in the inner shelf and on Georges Bank as a result of the tidal mixing

and continuous nutrient supply (Bias: chlorophyll-a (with DOM) = 4 mg chl m$^{-3}$, chlorophyll-a (without DOM) = 6

mg chl m$^{-3}$).

**3.2 Macronutrients (N, P, S)**

Nutrients such as nitrate, phosphorous, and silica have been well represented by several biogeochemical models

(Aumont et al., 2015; Das et al., 2019; Dutkiewicz et al., 2015; Jung et al., 2019b; Lachkar et al., 2019; Pant et al.,

2018; Le Quéré et al., 2016; Sankar et al., 2018). The PFTs based models such as PISCES-NEMO, DARWIN-

MITgcm, and TOPAZ-NEMO have shown well representation of surface nutrients distribution in the global oceans

with r (> 0.9 for P and N; ~0.85 for S), > 0.9 for all nutrients, and > 0.95 for all nutrients respectively. Both TOPAZ

and PISCES have represented: (i)  similar distribution of nutrients over global oceans, (ii) an overestimation in the

Southern Pacific Ocean bias of ≥ 4.5 μmol kg$^{-1}$, ~0.32 μmol kg$^{-1}$, and ≥ 16 μmol kg$^{-1}$ for nitrate, phosphate, and

silicate respectively , (iii) higher positive bias of nitrate (≥ 0.16 μmol kg$^{-1}$) and silicate (≥ 8 μmol kg$^{-1}$) in the central

and southern Pacific, and the Southern Ocean and (iv) underprediction of phosphate (bias ~-0.8 μmol kg$^{-1}$) at the

middle and higher latitudes in the Northern Hemisphere (Jung et al., 2019). The discrepancies in both models can be

attributed to the low resolution, weak North Atlantic deep waters, and strong ventilation in the Antarctica waters.

However, the improvement of the optical constituents by increasing the absorption of the optical constituents resulting

in a reduction in the size of oligotrophic regions in the subtropical gyres could be a solution as proposed by

(Dutkiewicz et al., 2015) using DARWIN-MITgcm. This has led to an enhancement of lateral nutrients supplies caused

by a decrease of productivity in high latitude. Furthermore, skill assessment of 21 regional and global coupled

biogeochemical models based on functional group approach including (PISCES, PlankTOM, COBALT, TOPAZ,

HAMOCC, BIOMASS, MEDUSA, ERSEM, PELAGOS, PISCES, NOBM) were conducted for the Arctic region

studies in order to investigate the capability of these models in representing the observed nitrate, mixed layer depth,

as well as euphotic layer depth. Most of the models have shown positive bias for the depth averaged nitrate explaining

the overestimation of nitrate in the upper layer (r ≤ 0.68) and none of these models were able to well represent the



variability in the field measurements. However, REcoM is applied and has shown to have a good performance for
DIN and silicate (r = 0.75) when coupled to a high resolution setup in the Arctic regions (Schourup-Kristensen et al.,

395   2018).

**3.3 Micronutrients (Fe)**

Low iron concentrations were simulated in the North and North Central Pacific, North Atlantic, and Antarctic whereas
in the North and Equatorial Indian and North Central Atlantic high levels were predicted using NOBM. The
overestimation has resulted in r ~ 0.86 and the reason of the iron overprediction is attributed to the lack of scavenging,
excessive remineralization, and slow detritus sinking rate. However, PISCES-NEMO  has shown a significant
underestimation for the spatial variability of iron in the global ocean with r = 0.75 suggesting a need to increase or
make the sediment source of iron highly variable since it is too small in the model (Aumont et al., 2015). It is also
suggested that iron supply to the surface layer is highly driven by eddies using a simplified version of DARWIN
biogeochemical model of two species as described in (Dutkiewicz et al., 2009) coupled to a flat bottom zonally re-
entrant  MITgcm model (Uchida et al., 2019). So, a better representation of the iron fluxes in the Southern Ocean
requires correct energetics of the mesoscale field which can be done by resolving and parametrizing the inverse energy
cascade caused by baroclinic instabilities of meso and sub-mesoscale (Person et al., 2019). Hence, (Jiang et al., 2019)
applied a modified version of Chai model coupled to ROMS which involved two phytoplankton groups (small
phytoplankton and diatoms), two zooplankton groups (micro and meso zooplankton), nutrients (nitrogen, silicate,
iron) indicating that dominant iron sources in the Scotia Sea are derived from sediments in the Antarctic Peninsula
shelf along with the South Orkney Plateau. In addition to these sources, the Antarctic Circumpolar Current, the
northern side of the Weddle Gyre, upwelling, atmospheric dust deposition, and icebergs are the common sources of
iron in the Southern Ocean (Jiang et al., 2019). The iron levels estimated by the modified Chai model have shown an
average overestimation by 0.26 nM deviated from the observed average value of 0.35 nM resulting in r = 0.76.
**3.4 Carbon**

Ocean carbon has been derived in different forms including the particulate organic and inorganic carbon (POC,
PIC),  partial pressure of $CO_2$ (pCO2) and dissolved inorganic carbon (DIC). A representation of the variable POC
reactivity evolved from reactive continuum model suggested by (Boudreau and Ruddick, 1991) and was introduced



in PISCES with a coarse resolution NEMO for the global oceans (Aumont et al., 2017). With the POC introduction
into the model, the POC levels in the ocean's interior increased by 1 to 2 orders of magnitude which has resulted from
the slow sinking of small particles from the surface. In addition, an increase of higher than a factor of 2 of the carbon
reaching the sediments has been attained showing better agreement with observations with Root Mean Squared Error
(RMSE) of 0.14 (without continuum reactivity), and 0.08 (with continuum reactivity). In addition, PISCES-MITgcm
has been used to qualitatively study the carbon cycle in the Arctic Ocean showing the capability in capturing the
observed seasonal and regional trends of the dissolved pCO2 (Manizza et al., 2011). However, the spring surface
pCO2 in the Canadian Archipelago is underestimated (~300 µatm) relative to observations (400-450 µatm) but able
to capture the summer levels (200-250 µatm). The riverine POC and the impact of terrestrial carbon resulted from
coastal erosion were neglected in the model hence caused the underprediction of carbon balance. Additionally, the
sedimentation and resuspension processes were neglected by the model which may be important in the enrichment of
the water column with carbon hence impacting air-sea gas exchange. In addition, the modelled surface DIC by
PISCES-NEMO was comparable to observations with r of 0.91 (Aumont et al., 2015). With regard to the comparison
conducted between PISCES-NEMO and TOPAZ-NEMO (Jung et al., 2019) for the surface DIC, TOPAZ-NEMO has
represented similar agreement with observation (r > 0.95), and the zonal averaged surface content is better represented
by TOPAZ-NEMO in the Northern Hemisphere (bias < 10 µmol kg$^{-1}$). Similarly, compared to observations lower
bias is shown for surface alkalinity by TOPAZ-NEMO than that of PISCES-NEMO in all oceans (e.g. Southern
Hemisphere: negative bias ~ 80 µgmol kg$^{-1}$, Equator: positive bias, mainly ≤ 16 µmol kg$^{-1}$, Indian Ocean: negative
bias, mainly < 32 µmol kg$^{-1}$, Pacific: positive bias, mainly ≤ 16 µmol kg$^{-1}$, Atlantic: negative bias, mainly < 64 µmol
kg$^{-1}$)
**3.5 Oxygen**

Surface oxygen level was estimated in the global oceans by PISCES-NEMO which has resolved the oxygen
distribution with r ~ 0.97 because oxygen reach closely to its solubility level and hence is constrained by sea surface
temperature (Aumont et al., 2015). Moreover, TOPAZ and PISCES coupled to NEMO (Jung et al., 2019) have shown
comparable spatial distributions (r ~0.98) and zonal averages of surface dissolved oxygen (DO) to the observed data
for DO in the global oceans but the overall was underestimated by TOPAZ (bias ~ -10 µmol kg$^{-1}$) and a slight



overprediction of DO (bias ~10 μmol kg$^{-1}$) was observed by PSICES except in polar regions. Because in polar regions
(areas at 60°N or higher) the availability and quality of satellite data is limited.
Both models have shown negative bias (~ 25 μmol kg$^{-1}$) in deep waters which was caused by the weak North Atlantic
deep waters represented by the physical model. TOPAZ-NEMO has also shown a better representation of the oxygen
minimum zone in the Pacific Ocean.  ERSEM coupled to a 1D Princeton/Mellor–Yamada and GOTM physical models
respectively were also implemented to study the global oxygen minimum zone. ERSEM-Princeton/Mellor–Yamada
lacks horizontal advective processes which could be enhanced through considering the diurnal physical processes
while ERSEM-GOTM ignores the episodic intrusion of oxygen within the oxygen minimum zone (Sankar et al., 2018)
(Sankar et al., 2018) (Blackford and Burkill, 2002). Thus, the models have shown contradictions between the estimates
and climatological seasonal cycles of oxygen at depths which might be attributed to the lack of lateral circulation in
the model. Further studies on the oxygen minimum zone in the Arabian Sea were conducted by (Lachkar et al., 2017,
2019, 2020) using Fasham model coupled to ROMS indicating that the primary productivity and oxygen minimum
zone are highly impacted by monsoon wind intensification with an overall high  r ~ 0.93 for oxygen (both seasons) in
the upper layer. Nevertheless, the model was incapable of resolving the full eddy spectrum because the resolution was
overly coarse. The model has considered the nitrogen as a limiting nutrient neglecting iron, phosphate, and silicate
which are the major nutrients limiting phytoplankton growth which may have led to amplify the impact of
denitrification on the nitrogen budgets in the Arabian Sea (Lachkar et al., 2017) thus overpredicting the oxygen
minimum zone.

**5 Conclusions**

This review presents the common biogeochemical models applied on various ecosystems of the world's ocean. These
models are evaluated through reviewing the studies that have been conducted to estimate biochemical parameters such
as chlorophyll-a, nutrients as well as carbon and oxygen components. Therefore, applications of biogeochemical
models on different ecosystems have shown different performances depending on the complexity of these ecosystems
and the governing equations. PFT's model approach has proven to be a good estimate of surface nutrients such as
nitrogen (N), phosphorous (P), and silica (S) in global oceans with r $\geq$ 0.85, $\geq$ 0.9, and $\geq$ 0.78 with some



inconsistencies apparent if coupled with the low-resolution physical component. NPZD models, for example Fasham,
are capable of accurately estimating N, P, and oxygen (O) with $R^2 > 0.9$ (for N, P), and $> 0.9$ (for O) in the Indian and
Pacific ecosystems. In contrast, the most effective prediction of iron with r is obtained between 0.7 and 0.86,
particularly for models using the functional group approach. In comparison, the reported performance for chlorophyll-
a varies between models and r can range from 0.55 to 0.9. These varying reported performances for these
biogeochemical parameters are dependent on the features of the ecosystems and reliability of the physical model.
Therefore, when developing the biogeochemical model, it is necessary to take into consideration the most appropriate
physical models.
**6 Author contribution**
Kaltham Ismail and Maryam R. Al Shehhi defined the content of the manuscript. Kaltham Ismail prepared the
manuscript and Maryam R. Al Shehhi contributed in revisions and improvements.
**7 Competing interests**
The authors declare no conflict of interest.
**8 Acknowledgements**
The authors would like to thank Khalifa University for the financial support. This paper is under the project of
modelling the biogeochemistry of the Arabian Gulf waters.









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



Table 1. Typical functional forms of phytoplankton response to irradiance I. These functional forms can be multiplied
by the maximum photosynthesis rate termed as $P_{max}$ in some processes. Adapted from (Franks P, 2002).

| Functional Form | Definition |
|---|---|
| $\dfrac{I}{I_o}$ | Linear response to incident light |
| $\dfrac{I}{I_o + I}$ | Saturating response |
| $1 - \exp\left(-\dfrac{I}{I_o}\right)$ | Saturating response |
| $\tanh\left(-\dfrac{I}{I_o}\right)$ | Saturating response |
| $\dfrac{I}{I_o}\exp\left(1 - \dfrac{I}{I_o}\right)$ | Saturating and photo-inhibiting response. $I_o$ represent the maximum photosynthesis rate. |


Table 2. Some of the commonly used functional forms of phytoplankton nutrient uptake. Adapted from (Leles et al.,

781 2016)

| Functional Form* | Description |
|---|---|
| $\dfrac{V_{max} \cdot N}{k + N}$ | Michaelis-Menten/Monod; hyperbolic |
| $\mu_{max}\dfrac{1 - \dfrac{Q_{min}}{Q}}{1 - \dfrac{Q_{min}}{Q_{max}}}$ | Quota Curve; hyperbolic |
| $\mu_{max}\dfrac{Q - Q_{min}}{(Q - Q_{min}) + K}$ | Quota Curve; rectangular- hyperbolic |
| $\mu_{max}\dfrac{(1 + KQ) \cdot (Q - Q_{min})}{(Q - Q_{min}) + KQ \cdot (Q_{max} - Q_{min})}$ | Normalized Quota equation; rectangular-hyperbolic |

* $V_{max}$: maximum utilization rate of nutrient; $k$: affinity constant for nutrient uptake; $Q_{min}$: minimum nutrient quota;
$Q_{max}$: maximum nutrient quota; $Q$: nutrient quota; $K$: half saturation constant for quota curve.








Table 3. some of the commonly used functional forms of phytoplankton zooplankton grazing. Adapted from: (Leles
et al., 2016)

| Functional Form* | Description |
|---|---|
| $a \cdot P$ | Holling Type I: linear |
| $\dfrac{a \cdot P}{1 + a \cdot h \cdot P}$ | Holling Type II: Hyperbolic |
| $\dfrac{I_{max} \cdot P}{k_1 + P}$ | Michaelis-Menten/Monod Hyperbolic |
| $\dfrac{a \cdot P^2}{1 + a \cdot h \cdot P^2}$ | Type III: sigmoidal |
| $\dfrac{I_{max} \cdot P^2}{k_1^2 \cdot P^2}$ | Michaelis-Menten/Monod Sigmoidal |
| $\dfrac{a_i \cdot P_i}{1 + \sum_1^n a_r \cdot h_r \cdot P_r}$ where n is the number of preys | Type II; hyperbolic. multiple preys |
| $\dfrac{I_{max} \cdot \sum C_{pi}}{k_1 + \sum C_{pi}}$ where $C_{pi} = C_{ri} \cdot P_i$ | Monod; hyperbolic; prey selectivity |

*$a$: attack rate; P = phytoplankton availability; h = handling time; $I_{max}$: maximum ingestion rate; $k_1$: half saturation
constant for ingestion; i: subscript for prey type; r = subscript relative to prey type weight; $C_{pi}$: potential capture rate;
$C_{ri}$: capture rate.

Table 4. Some of the functional forms for mortality rate of both phytoplankton and zooplankton. Adapted from:
(Franks P, 2002; Leles et al., 2016)

| Functional form of i(P) | Description |
|---|---|
| $m$ | Linear |
| $mP$ | Quadratic non-linear |
| **Functional form of j(Z)** | **Description** |
| $m \cdot Z$ | Linear |
| $\dfrac{m \cdot Z}{k_2 + Z}$ | Hyperbolic |
| $\dfrac{m \cdot Z^2}{k_2^2 + Z^2}$ | Sigmoidal |
| $\dfrac{m \cdot Z \cdot c}{k_2 + Z \cdot c}$ | Hyperbolic-intraguild predation |

$m$: mortality rate of zooplankton; $k_2$: half saturation constant for zooplankton closure term; $c$: zooplankton fraction
foe which closure terms acts.





Table 5. Biochemical models applied in the global ocean ecosystems including performance, physical
model and model resolution.

| Ocean | Model approach | Resolution (grid size) | Key biochemical variables [a] | Physical model | Performance[b] | Ref. |
|---|---|---|---|---|---|---|
| Global | Moore | 2-D global grid (100 X 116 grid-points); Longitudinal resolution of 3.6° and variable latitudinal resolution from 1–2° with higher resolution near the equator | Fe | NCAR | Bias: Fe = 265 pM (July-North Pacific) Fe = 14 pM (June-Equatorial Pacific) Fe = 121 pM (May-North Atlantic) Fe = 4514 pM (September/August-Arabian Sea) Fe = 2362 pM (November-Southern Ocean) | (Moore et al., 2001) |
| Global | NOBM | 3-D; 2/3° latitude and 1.25° longitude with 14 layers | Fe, Chl-a | GCM | Fe: r = 0.86 & $R^2$= 0.74 chl-a: r > 0.7 | (Gregg et al., 2003) |
| Global | HAMOCC5 | 3-D; horizontal resolution is uniformly 3.5 by 3.5 degrees with 22 vertical layers | Fe, Chl-a | LSG | Bias: Fe = 0.15 nM (at depth of 3000 m) chl-a = 0.24mg/m$^3$ | (Aumont et al., 2003) |
| Global | ERSEM | 1-D water column | Chl-a | GOTM | - | (Blackford et al., 2004) |
| Global | Moore | 3-D 100 X 116 horizontal grid points with a resolution of 3.6° longitude and 0.9°-2° latitude | Fe | CCSM | Refer to Moore et al., 2001 | (Moore et al., 2004) |
| Global | Moore | 3-D; 3.6° in longitude; 0.9°– 2° degree in latitude | IC, Fe | NCAR CCSM3 | Refer to Moore et al., 2001 | (Moore et al., 2006) |
| Global | Moore | 3-D; 3.6° in longitude and 0.8° to 1.8° latitude and 25 levels in the vertical | Fe | CCSM | Refer to Moore et al., 2001 | (Moore & Doney, 2007) |



| Global | PlankTOM | 3-D; 2° in longitude, 1.1° average in latitude with 31 vertical levels | Chl-a | NEMO | RSS = -13% | (Buitenhuis et al., 2013) |
|---|---|---|---|---|---|---|
| Global | PISCES | 3-D; 2° by 2°cosΦ (where Φ is the latitude) with a focusing of the meridional resolution to 0.5° in the equatorial domain. 30 vertical layers | IC, P, N, Fe, Alk, O, S | NEMO | r:<br>C = 0.91<br>P = ~0.91<br>N = 0.95<br>Fe = 0.75<br>Alk = ~0.8<br>O = 0.97<br>S = ~0.85 | (Aumont et al., 2015) |
| Global | DARWIN | 3-D; horizontal resolution of 1º x 1º with 23 levels | P, N, S, Chl-a | MITgcm | r:<br>P,N,S > 0.9<br>Chl-a ~ 0.55 | (Dutkiewicz et al., 2015) |
| Global | PlankTOM | 3-D; zonal resolution of 2° and a meridional resolution of 2°×cos(latitude) with 30 z levels | O | NEMO | - | (Andrews et al., 2017) |
| Global | PISCES | 3D; 2° by 2°cos(φ) (where φ is the latitude) with an increased meridional resolution to 0.5° in the equatorial domain. 30 vertical layers | POC | NEMO | RMSE:<br>No Reactivity Continuum (RC) = 0.14<br>With RC = 0.08 | (Aumont et al., 2017) |
| Global | Moore | 2-D; horizontal resolution of 0.27°–0.53° | S, N, P, O, DIC flux, Chl-a | NCAR-CSM1 | r:<br>S = 0.8<br>N = 0.95<br>P = 0.92<br>O = 0.85<br>DIC = 0.75<br>Chl-a = 0.6<br>(Doney et al., 2009) | (Pant et al., 2018) |
| Global | Moore | 3D; 60 vertical levels, was run at the nominal one-degree resolution | IC & coccolithophores | CESM | Refer to Moore et al., 2001 | (Krumhardt et al., 2019) |
| Global | TOPAZ & PISCES | 3-D; horizontal resolution of 2° × 2° (182 × 149 grid points) and meridional resolution of 0.5° with 31 levels | Chl-a, N, P, S, O, IC, Alk | NEMO | r:<br>chl-a :0.6-0.9 (both models)<br>N, P, O, & S, DIC & Alk<br>> 0.95<br>(Both models) | (Jung et al., 2019a) |





| | | | | | | |
|---|---|---|---|---|---|---|
| Atlantic | Fasham | 3-D; 2° horizontal resolution and 25 vertical levels | Chl-a, nutrients | MOM | RMSD = 0.97 (for detrital sinking rate of 10 m d⁻¹) & 0.77 (detrital sinking rate of 1 m d⁻¹) Based on Fasham et al, 1990 metrics | (Oschlies & Garçon, 1999) |
| Atlantic | Fasham | 3-D; horizontal resolution is 10 km, and 30 sigma levels | Chl-a | ROMS | Chl-a Winter: r = 0.75 Spring: r = 0.72 Summer: r = 0.85 Fall: r = 0.83 | (Fennel et al., 2008) |
| Atlantic | Fasham | 3-D; 10-km horizontal resolution and 30 terrain-following vertical levels | Chl-a | ROMS | Bias: Chl-a (with DOM) = 4 mgchl/m³ Chl-a (without DOM) = 6 mgchl/m³ | (Druon et al., 2010) |
| Atlantic | Fasham | 3-D; horizontal resolution of 5 km, and 36 vertical terrain-following layers | Chl-a | ROMS | r: chl-a = Spring: 0.6 Summer: 0.65 Fall: 0.53 Winter: 0.45 | (Xue et al., 2013) |
| Atlantic | ERSEM | 3-D; 1/8° horizontal resolution with 43 vertical levels | P, N | OGCM-MED16 | r > 0.6 | (Lazzari et al., 2016) |
| Atlantic | ERSEM | 3-D; resolution of 1/10° X 1/10° (~10 X10 Km) in the horizontal axis and 24 sigma-levels in the vertical axis | Chl-a, P,N | POM | r: chl-a = 0.64 P = 0.02 mmolP/m³ N = 0.55 mmolN/m³ | (Kalaroni et al., 2019) |
| Atlantic | ERSEM | 3-D; resolution of 1/10° x 1/10° (~10 X10 Km) in the horizontal axis and 24 sigma-levels in the vertical axis | Chl-a, P | POM | Refer to Kalaroni et al., 2019 | (Kalaroni et al., 2020) |
| Indian | Fasham | 3-D; 1° resolution in latitude and longitude with 10 vertical levels | Chl-a, N | OGCM | Bias: Chl-a = 0.1 N = -11 mmolN/m³ | (Ryabchenko et al., 1998) |
| Indian | ERSEM | 1-D; grid size of approximately 20 km with 40 vertical layers | N | Princeton/Mellor –Yamada | Bias: N = 9 mmol/m³ | (Blackford & Burkill, 2002) |


| | | | | | Bias: | |
|---|---|---|---|---|---|---|
| Indian | McCreary | 1-D; with 4 vertical layers | Chl-a, N | Four-layer model | Chl-a = 2 mg chl-a/m$^3$ N = 5 molN/kg | (Hood et al., 2003) |
| Indian | Fasham | 3-D; horizontal resolution of 1/3º both meridionally and zonally with 35 levels | Chl-a, N | MOM | Bias: Chl-a = 0.40 (scale: 0-2 mg/m$^3$) N = 15 (depth of 75 m) (0-30 mmolN/m$^3$) | (Kawamiya & Oschlies, 2003) |
| Indian | PISCES | 3-D; mean horizontal resolution of 0.5° by 0.5° cos $\phi$ (where $\phi$ is the latitude) with 30 vertical layers | Chl-a | NEMO | - | (Koné et al., 2009) |
| Indian | PISCES | 3-D; resolution 1/12° (~9 km) horizontal grid with 46 vertical layers | Chl-a, Fe | NEMO | Bias: Fe = 0.15 nM (at depth of 3000 m) chl-a = 0.24mg/m$^3$ | (Resplandy et al., 2011) |
| Indian | McCreary | 1-D; 6 vertical layers | Chl-a | Six-layer model | Bias: Chl-a = 2 mgchl-a/m$^3$ | (McCreary et al., 2013) |
| Indian | Fasham | 3-D; 1/12º horizontal resolution with 32 vertical sigma layers | Chl-a, N, O | ROMS | Chl-a: r between 0.69 (summer);0.74(winter) N: r = 0.88 O: r = 0.93 | (Lachkar et al., 2017) |
| Indian | ERSEM | 1-D; 100 vertical levels | S, P, N, O | GOTM | r > 0.9 for S,P,N,O | (Sankar et al., 2018) |
| Indian | Fasham | 3-D; horizontal resolution of 1/24º and a vertical grid made of 32 levels | Chl-a, N, O | ROMS | Chl-a: r between 0.69-0.74 N: r = 0.88 O: r = 0.93 | (Lachkar et al., 2019) |
| Indian | PISCES | 3-D; grid resolution of 1/10° with 32 vertical layers | Fe | ROMS | Refer to Aumont et al., 2015 | (Guieu et al., 2019) |
| Indian | NOBM | 3-D; 1.25° longitude by 2/3° latitude with 14 vertical layers | N, S, Chl-a | OGCM | r: N = 0.9-0.96 S ~ 0.95 Chl-a = 0.78 (in situ) & 0.618 (SeaWiFS) | (Das et al., 2019) |



| | | | | | | |
|---|---|---|---|---|---|---|
| Indian | TOPAZ | 3D; 1° horizontal resolution with 1/3° resolution near the equator; 50 vertical layers | Fe | MOM | Bias:<br><br>Fe = 1.5 nMol/m³ | (Sharada et al., 2020) |
| Indian | Fasham | 3-D; 1/10∘ horizontal resolution with 32 sigma-coordinate vertical layers | Chl-a, N, O | ROMS | r:<br><br>N & O = 0.9; Chl-a = 0.42 (winter) ,0.67 (fall) | (Lachkar et al., 2020) |
| Southern | PlankTOM | 3-D; 2∘ of longitude and a mean resolution of 1.5∘ of latitude with 30 vertical levels | Chl-a, N, S, P | NEMO | r:<br>Chl-a ~ 0.8(PlankTOM6)<br>Chl-a ~ 0.81(PlankTOM10)<br>P ~ 0.9 (PlankTOM6)<br>P ~ 0.92(PlankTOM10)<br>N ~ 0.9 (PlankTOM6)<br>N ~ 0.85(PlankTOM10)<br>S ~ 0.8(PlankTOM6)<br>S ~ 0.78(PlankTOM10) | (Le Quéré et al., 2016) |
| Southern | NOBM | 3-D; 2/3º latitude and 1 ¼ ºlongitude with 14 vertical layers | PIC,N, S | OGCM | - | (Trull et al., 2018) |
| Southern | DARWIN | 3D; three horizontal grid spacings are used: 20, 5, and 1 km with 76 vertical layers | Fe | MITgcm | No detailed skill analysis of the biological state variables against observations | (Uchida et al., 2019) |
| Southern | PISCES | 3-D; 1∘ horizontal resolution on an isotropic mercator grid with a local meridional refinement up to 1/3∘ at the Equator with 75 levels | Fe | NEMO | Refer to Aumont et al., 2015 | (Person et al., 2019) |
| Sothern | DARWIN | 3-D; with mean horizontal spacing of 18 km and 50 vertical levels | Chl-a | MITgcm | MAE:<br>0.74 mg chl-a m⁻³(diatoms)<br>0.22 mg chl-a m⁻³(haptophytes) | (Lo et al., 2019) |
| Southern | Chai | 3-D; horizontal scale of 2-3 km with 40 vertical layers | Fe | ROMS | r = 0.76 | (Jiang et al., 2019) |
| Southern | TOPAZ | 3-D; 1° × 1° horizontal resolution with increased resolution near the Equator and 50 unevenly spaced vertical levels in depth coordinates | DIC, N | ESM2M | - | (Bronselaer et al., 2020) |





| | | | | | | |
|---|---|---|---|---|---|---|
| Arctic | PISCES | 3-D; horizontal resolution with an average spacing of ~ 18 km and 50 levels | DIC | MITgcm | No detailed skill analysis available | (Manizza et al., 2011) |
| Arctic | 21 coupled biogeochemical models with different physical systems [c] | - | Majority includes N, P, S, Fe | - | r: see below the table [d] | (Babin et al., 2016) |
| Arctic | REcoM2 | 3-D; resolution north of 60°N equals 4.5 km, between 40 and 60°N it is approximately 25 km, while a resolution of nominal 1° is used south of 40°N; 32 vertical levels | DIN, S, Chl-a | FESOM | r: DIN: 0.75 Si:0.75 Chl-a: 0.56 Bias: chl-a ≤ 0.1 mg/m$^3$; (Schourup-Kristensen et al., 2014) | (Schourup-Kristensen et al., 2018) |
| Arctic | BLING | 3-D; horizontal resolution of 0.25° with 50 vertical levels | Chl-a, DIM | NEMO | $R^2$: IC ≥ 0.93; Chl-a ≥ 0.76 except in BB & HB regions where $R^2$ = 0.1 & 0.4 respectively; DIM: 0.84, 0.82, 0.93 for BS, LS, BG respectively. Yet $R^2$ = -0.21 in BB | (Castro de la Guardia et al., 2019) |
| Arctic | DARWIN | 3-D; 18 km of horizontal resolution with 50 vertical levels | DIC | MITgcm | No detailed skill analysis available | (Manizza, 2019) |
| Pacific | Leonard | 1-D vertical ecosystem model; latitudinal resolution of (1/3)° near the equator | Chl-a | OGCM | r: chl-a = 0.55 & 0.93 if data from June-August 1998 are excluded | (Christian et al., 2001) |
| Pacific | Chai | 3-D; horizontal resolution of 1/8 degree with 30 levels in the vertical direction | Chl-a | ROMS | Bias: Chl-a = 0.18 mg/m$^3$ (Scale: 0.05-0.4) | (Xiu & Chai, 2011) |



| Pacific | Fasham | 3-D; 3 km horizontal grid size with 30 vertical levels | N, P | ROMS | RMSD = 0.97 (for detrital sinking rate of 10 m d$^{-1}$) & 0.77 (detrital sinking rate of 1 m d$^{-1}$) based on (Fasham et al., 1990) | (Gan et al., 2014) |
| Pacific | Fasham | 3-D; $(1/12)° \times (1/12)°$ of horizontal resolution, 5 d of temporal resolution and 22 sigma levels | O, N | ROMS | $R^2$: O = 0.88 N = 0.95 | (Ji et al., 2017) |
| Pacific | PISCES | 3D; resolution of 7.5 km and 32 sigma levels | Chl-a, Fe | ROMS | Bias: Chl-a = 12 mg/m$^3$ Fe = 2.5 nM | (Vergara et al., 2017) |
| Pacific | TOPAZ | 1-D; single water column | Chl-a, O, N, P, S, CO$_2$ | GOTM | r: chl-a = 0.53 O = 0.47 N = 0.31 P = 0.16 S = 0.19 CO$_2$ = 0.94 | (Jung et al., 2019b) |
| Pacific | Chai | 1D; 100 layers in the vertical direction | Chl-a | ROMS | r > 0.6 | (Ma et al., 2019) |
| Pacific | Fasham | 3-D; horizontal resolution ranged from ~7 km in the north to ~10 km in the south with respect to a cylindrical map projection with 30 vertical levels | P & N | ROMS | $R^2$ > 0.9 | (Lu et al., 2020) |
| Pacific | Kearney | 3-D; 10km horizontal resolution with 30 depth levels | Chl-a | Bering 10K ROMS | No detailed skill analysis of the biological state variables against observations | (Kearney et al., 2020) |

a) **N**: nitrogen (NO$_3$, NH$_4$); **P**: phosphorus; **S**: silicon; **C**: carbon; **O**: oxygen; **Chl**: chlorophyll; **DIC**:
dissolved inorganic carbon; **PIC**: particulate inorganic carbon; **DIN**: dissolved inorganic nitrogen; **DIC**:
dissolved inorganic carbon; **POC**: particulate organic carbon; **Alk**: Alkalinity; **TP**: total phosphorous;
**TN**: total nitrogen; **NO**x: nitrate+nitrite; **DO**: dissolved oxygen.
b) **r**: Pearson correlation coefficient; **R$^2$**: coefficient of determination; **SCC** = Spearman correlation
coefficients; **RSS**: residual sum of squares; **RMSD/E**: root mean square difference/error; **MAE**: mean
absolute error.
c) 1. NEMO-PISCES; 2.NEMO-PlankTOM5.3; 3.NEMO-PlankTOM10; 4.MOM-COBALT; 5.MOM-
TOPAZ; 6.MICOM-HAMOCC; 7.POP-BIOMASS; 8.NEMO-MEDUSA; 9.NEMO-ERSEM; 10.NEMO-
updated configuration of ERSEM; 11.PELAGOS(NEMO-BFM); 12.POP-Moore; 13.NEMO-PISCES;
14.NEMO-PISCES; 15.NEMO-PISCES; 16.NorESM(HAMOCC); 17. GISS-E2-R-CC(NOBM); 18.MPI-
OM HAMOCC.
d) 0.46; -0.01; 0.28; 0.68; 0.66; -0.02; 0.05; 0.49; 0.27; 0.28; 0.57; 0.30; 0.46; 0.06; 0.56;










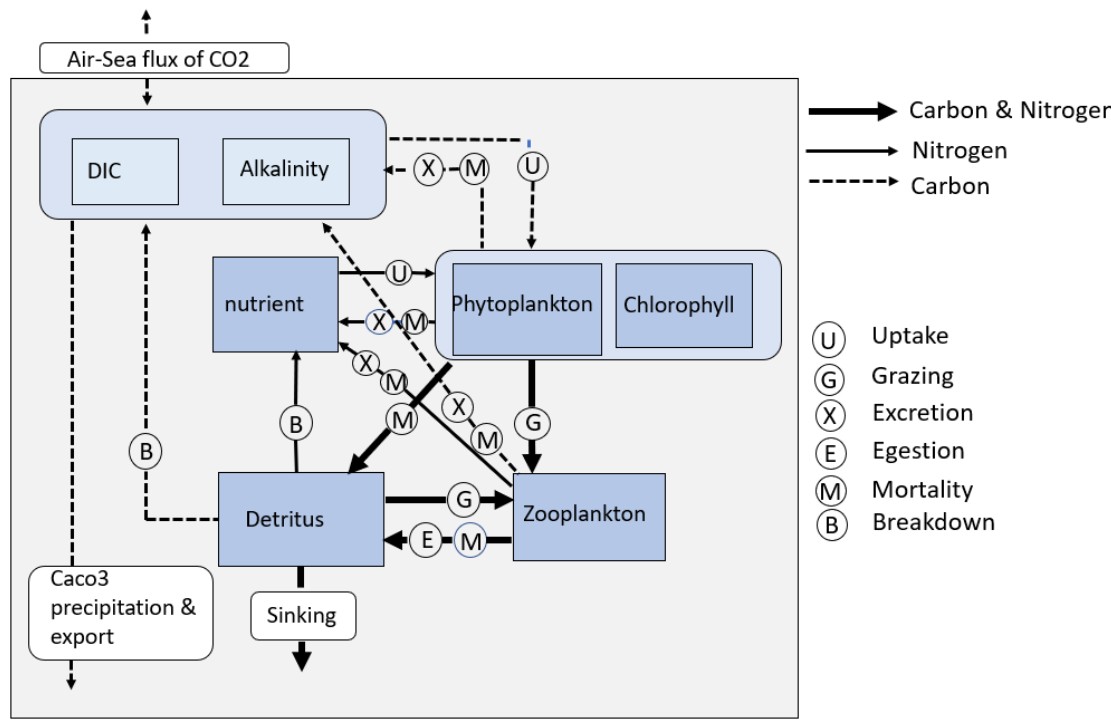


Figure 1. Schematic representation of the Hadley Centre Ocean Carbon model (HadOCC) which is an NPZD model
coupled with carbon cycle (Palmer & Totterdell, 2001). The labelled boxes nutrient, phytoplankton, zooplankton, and
detritus are representing the
