# Peer review of "Reviews and syntheses: Assessment of Biogeochemical Models in the Marine Environment"

_Biogeosciences, 2021_

## Referee Comment (RC1)

Review of "**Reviews and syntheses: Assessment of Biogeochemical Models in the Marine Environment**" by Kaltham A. Ismail and Maryam R. Al-Shehhi

**Summary**

The authors have written a review of marine biogeochemical models.  An updated review on this topic is highly needed and would be very useful for the community. The authors have structured the paper into an introduction, a section on the structure of biogeochemical models, and a section where they evaluate the performance of BGC models in various parts of the ocean. I think this structure would work well, however there are many important aspects, and details, of marine biogeochemical modeling that are lacking in the manuscript (please see my comments below for more details). Apart from my comments below, one additional major detail that needs to be thought about is whether the authors want to focus on a specific type of biogeochemical models (for instance, global, regional, or biogeochemical models used within the CMIP framework), or if they want to write a general review on biogeochemical modeling. This is not very clear in the current manuscript. Overall, the manuscript needs an overall major revision, both with respect to scientific content and writing/language, to reach the level that I would expect from a review paper on marine biogeochemical models.

**Major Comments**

**Introduction**

The introduction is difficult to read. There is no well defined structure (i.e., no red line) and the language needs some work (at the moment it is very much spoken language). Also, the scientific content needs to be worked on. In a review of marine biogeochemical modeling I would expect something like the following content:

1.  _Some introductory text on ocean biogeochemistry_, why it is important (for example for  global carbon cycle, marine ecosystems),  and its most important components. The authors have written some text on this on lines 28-36, but it is very short and not very informative. Try to make a link between the different components instead of just mentioning them one by one. Maybe a figure with an illustrative scheme could be useful.

2.  _Some information on the history of marine biogeochemical models_: The authors mention the model of Fasham et al., 1990. Some more references to possibly earlier and later models would be highly valuable, and how the models are connected through history. Maybe you could create a flow diagram showing how the models have developed? You could also mention their inclusion in ocean circulation models and Earth System Models.

3.  _Some examples on why these models are important_: You could mention some major findings that have been made by the use of marine biogeochemical models, and the role they play for example in assessments of the global carbon budget and in the IPCC reports.

**Section 2: Biogeochemical modelling approaches**
It is unclear to me why you have chosen these three types of classes for biogeochemical models, specifically? In my opinion the classes are not that distinct, and with this classification you are missing several important aspects of biogeochemical models. I would suggest you have the following subsections under section 2:

2.1 Classical NPZD models
I like that you start with this section. It is important as it explains the most simple biogeochemical model possible.

2.2 Adding more elements (other nutrients and carbon)
Here it would be interesting if you could describe the rationale behind including other elements, including carbon, phosphorus, iron, oxygen. You should describe that the most common way to represent the content of the different elements in plankton is by using the Redfield ratio, and what implications this has for the representation of biogeochemical cycling. You should also describe that there are models that use a flexible stoichiometry approach. In relation to this you can discuss the Droop-papers. Why do some models take into account chlorophyll?
Here (or in the introduction) you can also mention that all BGC models have not been developed from plankton models, and that HAMOCC initially was  a pure inorganic carbon cycle model and was utilised to evaluate both the 12C cycle and the ocean model residence time properties.

2.3 Increasing the biological complexity
Here you can describe the approaches used to represent  several types of plankton, i.e. functional types. What functional types are the most common ones to include? What traits do they have? In relation to this you should discuss the inclusion of silicate (and diatoms). It would also be useful to take up nitrogen fixers here, and the process of nitrogen fixation.

2.4 The addition of several classes of organic matter and bacteria
Many biogeochemical models have several classes of organic matter (of different lability). An overview of these would be useful.

2.5 Sediment interactions
It would be useful with some text on the representation of sediment biogeochemistry in a review paper on biogeochemical models. Some biogeochemical models lack a sediment model. Does this matter?

**Section 3: Determination of the biogeochemical parameters**
I have several remarks on this section:

1) The title is very vague. I would suggest something like "Applications of biogeochemical models".

2) The content of this section needs a major revision. In a review paper on marine biogeochemical models, I would expect a section discussing the ability of these models to represent marine biogeochemical cycles. This includes several aspects; i) spatial variations (both vertical and horizontal), ii) temporal (including seasonal, interannual, decadal) variations. You do not necessarily have to discuss this for all state variables, but you can pick out important examples from the literature. At the moment you are mostly discussing the models' abilities to simulate spatial (?) variations by mentioning correlation coefficients and bias, but this does not tell the reader much. Please be more precise in your description. For example, do the models reproduce the deep oxygen minimum zones, and the high chlorophyll concentrations in the North Atlantic? Do they simulate the observed interannual variations in seawater pCO2? You can divide the section into some major zones that you discuss, for example the tropical, subtropical, seasonally stratified seas/oceans/high nutrient-low chlorophyll zones/ etc… If you want to bring in examples from smaller seas, as you do, this could also be very useful, but please structure the text better. At the moment it is jumping forth and back between regions, making it difficult to follow.
You have to provide a deeper discussion on the fact that the performance of biogeochemical coupled to ocean circulation models highly depends on the performances of the physical models. Therefore, you should be careful when you compare different models with different biogeochemical structure, and also different physical models. If they have different physical models, you cannot attribute the differences to the biogeochemistry.
3) Within this section I would recommend you not to only discuss the performance of the biogeochemical models, but also their applications, i.e., some examples of what scientific questions that can be answered with these models. Some examples are process studies, future projections, near-time predictions and forecasts, and reconstructions. In your manuscript you do not mention paleo-oceanography and future projections. I think that these are important applications that should be mentioned. If you want to specialize your review on "present days", please state so clearly somewhere in the beginning of the manuscript.
4) In this section, I would also expect a discussion on major issues and uncertainties related to these models, and a paragraph on their future development.
5) Please carefully revise the language and your formulations.

**Equations**
I think that it is enough that you show equations 1-4 in your paper. These give the basic idea of biogeochemical models. For more complex formulations, you can refer to model description papers. In other words, you can remove equations 5-14.

**Figures**
I cannot see that you refer to figure 1?

**Table 5**
This table is very difficult to read. How do you determine what are key biogeochemical variables? Moreover, I do not think that it is fair to determine their performance just by mentioning numbers representing correlation or bias. It does not say much (see my

comments above). I would completely remove the column on performance, and rather include some figures showing the performance of various models.

Maybe you could consider having a table like Table 1 in Seferian et al., 2020..

**Minor Comments**

L11-12: Replace "Therefore, this review…" with "This review …"

L13: Replace "Then, applications of these …" with "Applications of these…"

L16: Replace "models based on functional group approach when coupled to high-resolution physical models" with "models with a functional group approach coupled to high-resolution physical models"

L17: With "good estimates of surface nutrients" I guess that you mean "good spatial distribution of surface nutrients"? Please clarify this in the text.

L19: I suggest to remove "suitable"

L17-20: why do you mention correlation coefficients for the functional group models, and coefficients of determination for the NPDZ models?

L16 & L19: On line 16 you write functional group models coupled to high resolution models, but on line 19 when you write about the NPZD models, you do not mention anything about the resolution. From this the reader get the impression that you only look into high resolution models in the first case, but not in the second. I would suggest just to remove the "high resolution"

L25-26: I do not agree with this: NPZD models are also commonly used for studying biogeochemical cycling. Rather, models with functional groups have been developed for questions more oriented towards ecosystems (Baird et al., 2022).

L39: Models that include fishes and whales are not biogeochemical, but ecological models. Please remove fishes and whales.

L127-140: This text does not fit under this subsection. I would suggest you to distribute it under the other subsections of section 2 that I suggested you to include.

L127-130: I get the impression that this part describes the introduction of nitrogen fixation, but you do not mention it explicitly.

L146: replace "which is a pure inorganic carbon" with "which initially was a pure inorganic carbon" (today HAMOCC includes biology)

L281-283: Are you sure that a better representation of chlorophyll is a result of the inclusion of several functional types? There are many other parameters that may differ between these models (as you for example write on L286-289)

L418-419: pCO2 is not a form of carbon, please remove

L418-421: I would suggest you to include this in the subsection on organic matter that I suggested you to include

References:

Baird, M., Dutkiewicz, S., Hickman, A., Mongin, M., Soja-Wozniak, M., Skerratt, J., & Wild-Allen, K. (2022). Modeling phytoplankton processes in multiple functional types. In *Advances in Phytoplankton Ecology* (pp. 245-264). Elsevier.

Séférian, R., Berthet, S., Yool, A. *et al*. Tracking Improvement in Simulated Marine Biogeochemistry Between CMIP5 and CMIP6. *Curr Clim Change Rep* 6, 95–119 (2020). https://doi.org/10.1007/s40641-020-00160-0

---

## Author Comment (AC1)

**Reply to reviewers' comments**

Title: "Reviews and syntheses: Assessment of Biogeochemical Models in the Marine Environment"

To the reviewer

*The authors would like to thank the reviewer for his useful comments that help to improve the manuscript. The authors have considered all suggestions and addressed the raised issues trying to provide necessary clarifications and improvements. Based on the reviewers' comments, the authors have started rewriting the three sections (introduction, model approaches, applications of biogeochemical models in the global oceans) as per the reviewer's suggestion and requests.*

*Below are given point by point answers to the comments. All the changes will be implemented in the revised manuscript.*

**Comments from the editors and reviewers:**

Reviewer Comments:
Reviewer 1
Summary
The authors have written a review of marine biogeochemical models. An updated review on this topic is highly needed and would be very useful for the community. The authors have structured the paper into an introduction, a section on the structure of biogeochemical models, and a section where they evaluate the performance of BGC models in various parts of the ocean. I think this structure would work well, however there are many important aspects, and details, of marine biogeochemical modeling that are lacking in the manuscript (please see my comments below for more details). Apart from my comments below, one additional major detail that needs to be thought about is whether the authors want to focus on a specific type of biogeochemical models (for instance, global, regional, or biogeochemical models used within the CMIP framework), or if they want to write a general review on biogeochemical modeling. This is not very clear in the current manuscript. Overall, the manuscript needs an overall major revision, both with respect to scientific content and writing/language, to reach the level that I would expect from a review paper on marine biogeochemical models.

*The authors have taken the reviewer's comments into consideration and they have clarified that the review focuses on the general biogeochemical modeling in the revised manuscript. The manuscript has been improved and will be further revised with more scientific discussions. The English will be revised by a native English speaker.*

Major Comments
Introduction
The introduction is difficult to read. There is no well-defined structure (i.e., no red line) and the language needs some work (at the moment it is very much spoken language). Also, the scientific content needs to be worked on. In a review of marine biogeochemical modeling, I would expect something like the following content: 1. Some introductory text on ocean biogeochemistry, why it is important (for example for global carbon cycle, marine ecosystems), and its most important components. The authors have written some text on this on lines 28-36, but it is very short and not very informative. Try to make a link between the different

components instead of just mentioning them one by one. Maybe a figure with an illustrative scheme could be useful.

*The authors have updated the introduction and started with an introduction on the ocean biogeochemistry. For example, the components of the ocean biogeochemical model have been introduced and the interactions between these components have been linked and discussed accordingly. These discussions have been also supported with a schematic diagram as shown below which will be further improved:*

[Figure]

Overview of the interactions of marine life in nutrient cycling

2. Some information on the history of marine biogeochemical models: The authors mention the model of Fasham et al., 1990. Some more references to possibly earlier and later models would be highly valuable, and how the models are connected through history. Maybe you could create a flow diagram showing how the models have developed? You could also mention their inclusion in ocean circulation models and Earth System Models. 3. Some examples on why these models are important: You could mention some major findings that have been made by the use of marine biogeochemical models, and the role they play for example in assessments of the global carbon budget and in the IPCC reports.

*The authors have considered the comments and have written about the history of biogeochemical models and the improvements in the modeling since then. The importance of the biogeochemical modeling has also been discussed. The flow diagram showing how the models have been developed is created as shown below. The figure will be further improved.*
*In addition, the inclusion of biogeochemical models into Earth System Models as well as its importance are added into the text. All of these changes are considered in the revised manuscript.*

[Figure]

Development of biogeochemical models starting with the pioneering work of Steel's (1958) to the more intricate model structure including plankton functional types.

Section 2:

Biogeochemical modelling approaches

It is unclear to me why you have chosen these three types of classes for biogeochemical models, specifically? In my opinion the classes are not that distinct, and with this classification you are missing several important aspects of biogeochemical models. I would suggest you have the following subsections under section 2:

2.1 Classical NPZD models

I like that you start with this section. It is important as it explains the most simple biogeochemical model possible.

2.2 Adding more elements (other nutrients and carbon)

Here it would be interesting if you could describe the rationale behind including other elements, including carbon, phosphorus, iron, oxygen. You should describe that the most common way to represent the content of the different elements in plankton is by using the Redfield ratio, and what implications this has for the representation of biogeochemical cycling. You should also describe that there are models that use a flexible stoichiometry approach. In relation to this you can discuss the Droop-papers. Why do some models take into account chlorophyll? Here (or in the introduction) you can also mention that all BGC models have not been developed from plankton models, and that HAMOCC initially was a pure inorganic carbon cycle model and was utilised to evaluate both the 12C cycle and the ocean model residence time properties.

*The authors would like to thank the reviewer for his valuable comments and suggestions. The authors follow the points suggested. The authors have changed the sub-sections and replaced them with the suggested structure. In addition, all the comments concerning the Redfield ratio, stoichiometry approach and Droop-papers have been considered and the discussions have been improved. The HAMOCC statement suggested has been also added in the introduction section.*

2.3 Increasing the biological complexity
Here you can describe the approaches used to represent several types of plankton, i.e. functional types. What functional types are the most common ones to include? What traits do they have? In relation to this you should discuss the inclusion of silicate (and diatoms). It would also be useful to take up nitrogen fixers here, and the process of nitrogen fixation.

*The authors have re-written this section and followed the reviewer's suggestion by adding the details of the functional types and traits. In addition, the diatoms and nitrogen fixation processes have been discussed.*

2.4 The addition of several classes of organic matter and bacteria
Many biogeochemical models have several classes of organic matter (of different lability). An overview of these would be useful.

*The authors have added this section as per the reviewer's suggestion including different forms of organic matters. For instance, the semi-labile dissolved organic carbon (DOC) and semi-labile dissolved organic nitrogen (DON) have been included in the manuscript.*

2.5 Sediment interactions It would be useful with some text on the representation of sediment biogeochemistry in a review paper on biogeochemical models. Some biogeochemical models lack a sediment model. Does this matter?

*The authors have added this section as per the reviewer's suggestion and they have discussed the different mechanisms for the transport of particles and solutes supported by biological activity.*

Section 3:
Determination of the biogeochemical parameters I have several remarks on this section:
   1) The title is very vague. I would suggest something like "Applications of biogeochemical models".

*The authors will replace the title with the title suggested by the reviewer "Applications of biogeochemical models".*

2) The content of this section needs a major revision. In a review paper on marine biogeochemical models, I would expect a section discussing the ability of these models to represent marine biogeochemical cycles. This includes several aspects: i) spatial variations (both vertical and horizontal), ii) temporal (including seasonal, interannual, decadal) variations. You do not necessarily have to discuss this for all state variables, but you can pick out important examples from the literature. At the moment you are mostly discussing the models' abilities to simulate spatial (?) variations by mentioning correlation coefficients and bias, but this does not tell the reader much. Please be more precise in your description. For example, do the models reproduce the deep oxygen minimum zones, and the high chlorophyll concentrations in the North Atlantic? Do they simulate the observed interannual variations in seawater pCO2? You can divide the section into some major zones that you discuss, for example the tropical, subtropical, seasonally stratified seas/oceans/high nutrient-low chlorophyll zones/ etc… If you want to bring in examples from smaller seas, as you do, this could also be very useful, but please structure the text better. At the moment it is jumping forth and back between regions, making it difficult to follow. You have to provide a deeper discussion on the fact that the performance of biogeochemical coupled to ocean circulation models highly depends on the performances of the physical models. Therefore, you should be careful when you compare

different models with different biogeochemical structure, and also different physical models. If they have different physical models, you cannot attribute the differences to the biogeochemistry.

> *The authors agree with the reviewer in which they have initially focused on the spatial variations. However, the authors have added thorough discussions on the major findings of the biogeochemical models applied in different zones. These major findings cover identifying the Oxygen Minimum Zones (OMZ) for example in the tropical zones of the Arabian Sea, in addition to estimating the algal blooms regions in the subtropical zones of the Atlantic Ocean. Moreover, the authors have highlighted the high nutrients and low chlorophyll (HNLC) conditions at the Southern Ocean, Equatorial Pacific and North Pacific. The discussions have included the ability of the models to capture these conditions in different regions and temporal scales.*

3) Within this section I would recommend you not to only discuss the performance of the biogeochemical models, but also their applications, i.e., some examples of what scientific questions that can be answered with these models. Some examples are process studies, future projections, near-time predictions and forecasts, and reconstructions. In your manuscript you do not mention paleo-oceanography and future projections. I think that these are important applications that should be mentioned. If you want to specialize your review on "present days", please state so clearly somewhere in the beginning of the manuscript. 4) In this section, I would also expect a discussion on major issues and uncertainties related to these models, and a paragraph on their future development. 5) Please carefully revise the language and your formulations.

> *The authors appreciate the reviewer's suggestions. Basically, the authors focus on the present days cases and this statement has been added in the introduction as suggested by the reviewer. To discuss the major issues and uncertainties, the authors have added a section on limitations and future suggestions.*

Equations
I think that it is enough that you show equations 1-4 in your paper. These give the basic idea of biogeochemical models. For more complex formulations, you can refer to model description papers. In other words, you can remove equations 5-14.

> *The authors have taken this point into consideration and removed equations 5-14.*

Figures
 I cannot see that you refer to figure 1?

> *It is added in the manuscript as suggested.*

Table 5
This table is very difficult to read. How do you determine what are key biogeochemical variables? Moreover, I do not think that it is fair to determine their performance just by mentioning numbers representing correlation or bias. It does not say much (see my comments above). I would completely

remove the column on performance, and rather include some figures showing the performance of various models. Maybe you could consider having a table like Table 1 in Seferian et al., 2020.

*The authors have considered the reviewer's comment by removing the performance column and adapting the style of Seferian et al., 2020. to improve the representation of the table. The table is added at the end of this document, which will be improved further. In addition, a figure showing the performance of different models to estimate the surface Chl-a is provided below.*

Minor Comments
L11-12: Replace "Therefore, this review…" with "This review …"

*The authors have taken this point into consideration and replaced it as suggested.*

L13: Replace "Then, applications of these …" with "Applications of these…"

*The authors have taken this point into consideration and replaced it as suggested.*

L16: Replace "models based on functional group approach when coupled to high-resolution physical models" with "models with a functional group approach coupled to high-resolution physical models

*The authors have taken this point into consideration and replaced it as suggested.*

L17: With "good estimates of surface nutrients" I guess that you mean "good spatial distribution of surface nutrients"? Please clarify this in the text.

*Yes, the authors mean good spatial distribution of surface nutrients. It is clarified in the manuscript.*

L19: I suggest to remove "suitable"

*The authors have removed "suitable" as suggested.*

L17-20: why do you mention correlation coefficients for the functional group models, and coefficients of determination for the NPDZ models?

*These statistical metrics have been extracted from the papers thus they vary per each case study.*

L16 & L19: On line 16 you write functional group models coupled to high resolution models, but on line 19 when you write about the NPZD models, you do not mention anything about the resolution. From this the reader get the impression that you only look into high resolution models in the first case, but not in the second. I would suggest just to remove the "high resolution"

*The authors have removed the "high resolution" as suggested.*

L25-26: I do not agree with this: NPZD models are also commonly used for studying biogeochemical cycling. Rather, models with functional groups have been developed for questions more oriented towards ecosystems (Baird et al., 2022).

*The authors agree with the reviewer and this comment has been taken into consideration and the text has been updated accordingly.*

L39: Models that include fishes and whales are not biogeochemical, but ecological models. Please remove fishes and whales.

*The authors have removed the text that include fishes and whales as part of biogeochemical model but added this statement: the elemental cycling is less regulated by higher trophic levels (fish and mammals) hence they are primarily considered separately ".*

L127-140: This text does not fit under this subsection. I would suggest you to distribute it under the other subsections of section 2 that I suggested you to include.

*The authors have moved the text to section 2.2 as per the reviewer suggestion.*

L127-130: I get the impression that this part describes the introduction of nitrogen fixation, but you do not mention it explicitly.

*The authors have added an explicit discussion on the nitrogen fixation in section 2.3.*

L146: replace "which is a pure inorganic carbon" with "which initially was a pure inorganic carbon" (today HAMOCC includes biology.

*This has been replaced as suggested.*

L281-283: Are you sure that a better representation of chlorophyll is a result of the inclusion of several functional types? There are many other parameters that may differ between these models (as you for example write on L286-289).

*The authors agree with the reviewer. Thus, they have reviewed this statement and added a clarification that there are other potential reasons for this improvement in Chlorophyll a such as the appropriate physical model.*

L418-419: pCO2 is not a form of carbon, please remove.

*The authors have removed the $pCO_2$.*

[Figure]

The coupled biogeochemical-physical models applied in the regional seas to reproduce the surface chlorophyll-*a* concentrations along with their statistical performance. Red color represents high statistical correlation (r> 0.8), orange color represents medium statistical correlation (0.5 < r < 0.8), and blue represent low statistical correlation (r <0.5).

Table 1. Overview of the biogeochemical models reviewed in this work showing the model approach, physical model, resolution, Nutrient/element cycling and the number of the living/non-living components.

| Ocean | Zone | Model approach | Physical model | Resolution | | Nutrient/element cycling | | | | | | Living/non-living components | | | | Ref. |
|---|---|---|---|---|---|---|---|---|---|---|---|---|---|---|---|---|
| | | | | Vertical layers | Horizontal resolution | Fe | N | P | Si | O$_2$ | C | #of phytoplankton | # of zooplankton | # of detritus | # of Bacteria | Ref. |
| Global | | NOBM | GCM | | | + | + | | + | | | 4 | 1 | 2 | | (Gregg et al., 2003) |
| Global | | HAMOCC5 | LSG | | | + | + | + | + | | | 2 | 2 | 1 | | (Aumont et al., 2003) |
| Global | | ERSEM | GOTM | | | + | + | + | + | + | + | 4 | 3 | 1 | 1 | (Blackford et al., 2004) |
| Global | | Moore | CCSM | | | + | + | + | + | + | + | 4 | 1 | 1 | | (Moore et al., 2004) |
| Global | | Moore | CCSM | | | + | + | + | + | + | + | 4 | 1 | 2 | | (Moore & Doney, 2007) |
| Global | | PlankTOM | NEMO | | | + | + | | + | + | + | 5 | 2 | 2 | | (Buitenhuis et al., 2013) |
| Global | | PISCES | NEMO | | | + | + | + | + | + | + | 2 | 2 | | | (Aumont et al., 2015) |
| Global | | DARWIN | MITgcm | | | | + | + | + | + | + | 9 | 2 | 2 | | (Dutkiewicz et al., 2015) |
| Global | | PlankTOM | NEMO | | | | + | | | + | + | 6 | 3 | | 1 | (Andrews et al., 2017) |
| Global | | PISCES | NEMO | | | + | + | + | + | | + | 2 | 2 | 2 | | (Aumont et al., 2017) |
| Global | | Moore | NCAR-CSM1 | | | + | + | + | + | + | + | 4 | 1 | 2 | | (Pant et al., 2018) |
| Global | | TOPAZ/PISCES | NEMO | | | + | + | + | + | + | + | TOPAZ:3 PISCES: 2 | PISCES:2 | | | (Jung et al., 2019a) |
| Atlantic | Tropical | Fasham | MOM | | | | + | | | | | 1 | 1 | 1 | | (Oschlies and Garçon, 1999) |
| Atlantic | Subtropical | Fasham | ROMS | | | | + | | | | + | 1 | 1 | 1 | | (Fennel et al., 2008) |
| Atlantic | Subtropical | Fasham | ROMS | | | | + | | | | + | 1 | 1 | 2 | | (Druon et al., 2010) |
| Atlantic | Subtropical | Fasham | ROMS | | | | + | | | | | 1 | 1 | 2 | | (Xue et al., 2013) |
| Atlantic | Subtropical | ERSEM | OGCM-MED16 | | | | + | + | + | + | | 4 | 4 | 1 | 1 | (Lazzari et al., 2016) |
| Atlantic | Subtropical | ERSEM | POM | | | | + | + | + | + | + | 4 | 3 | 2 | 1 | (Kalaroni et al., 2019) |
| Atlantic | Subtropical | ERSEM | POM | | | + | + | + | + | + | + | 4 | 3 | 2 | 1 | (Kalaroni et al., 2020) |
| Indian | Tropical | Fasham | OGCM | | | + | | | | | | 1 | 1 | 1 | 1 | (Ryabchenko et al., 1998) |

| Region | Type | Biogeochemical model | Physical model | | | | | | | | | | | | | | Reference |
|---|---|---|---|---|---|---|---|---|---|---|---|---|---|---|---|---|---|
| Indian | Tropical | ERSEM | Princeton/Mellor–Yamada | | | | | + | | + | | | | 4 | 2 | 2 | 1 | (Blackford and Burkill, 2002) |
| Indian | Tropical | McCreary | Four-layer model | | | | | + | | | | | | 1 | 1 | 1 | | (Hood et al., 2003) |
| Indian | Tropical | Fasham | MOM | | | | | + | | | | | | 1 | 1 | 1 | | (Kawamiya and Oschlies, 2003) |
| Indian | Tropical | PISCES | NEMO | | | | + | + | + | + | | | | 2 | 2 | 3 | | (Koné et al., 2009) |
| Indian | Tropical | PISCES | NEMO | | | | | + | | + | | | | 2 | 2 | 3 | | (Resplandy et al., 2011) |
| Indian | Tropical | McCreary | Six-layer model | | | | | + | | | + | | | 1 | 1 | 2 | | (McCreary et al., 2013) |
| Indian | Tropical | Fasham | ROMS | | | | | + | | | + | | | 1 | 1 | 1 | | (Lachkar et al., 2017) |
| Indian | Tropical | ERSEM | GOTM | | | | | + | + | + | + | | | 4 | 3 | 1 | 1 | (Sankar et al., 2018) |
| Indian | Tropical | Fasham | ROMS | | | | | + | | | + | | | 1 | 1 | 1 | | (Lachkar et al., 2019) |
| Indian | Tropical | PISCES | ROMS | | | | + | + | + | + | | | | 2 | 2 | 3 | | (Guieu et al., 2019) |
| Indian | Tropical | NOBM | OGCM | | | | | + | | | | | | 4 | 1 | 3 | | (Das et al., 2019) |
| Indian | Tropical | TOPAZ | MOM | | | | + | + | + | + | | | | 3 | | 2 | | (Sharada et al., 2020) |
| Indian | Tropical | Fasham | ROMS | | | | | + | | | + | | | 1 | 1 | 1 | | (Lachkar et al., 2020) |
| Southern | HNLC | PlankTOM | NEMO | | | | | + | + | + | + | + | | 6 | 3 | 3 | 1 | (Le Quéré et al., 2016) |
| Southern | HNLC | NOBM | OGCM | | | | | + | | + | | + | | 4 | 1 | 3 | | (Trull et al., 2018) |
| Southern | HNLC | DARWIN | MITgcm | | | | + | | | | | | | 2 | 2 | | | (Uchida et al., 2019) |
| Southern | HNLC | PISCES | NEMO | | | | + | + | + | + | | | | 2 | 2 | 2 | | (Person et al., 2019) |
| Sothern | HNLC | DARWIN | MITgcm | | | | | + | + | + | | + | | 6 | 2 | | | (Lo et al., 2019) |
| Southern | HNLC | Chai | ROMS | | | | + | + | | + | | | | 2 | 2 | 3 | 1 | (Jiang et al., 2019) |
| Southern | HNLC | TOPAZ | ESM2M | | | | | + | | | | + | | 3 | | 2 | | (Bronselaer et al., 2020) |
| Arctic | Seasonally stratified | PISCES | MITgcm | | | | | | | | | + | | 2 | 2 | 2 | | (Manizza et al., 2011) |
| Arctic | Seasonally stratified | 21 biogeochemical models [c] | - | | | | | | | | | | | | | | | (Babin et al., 2016) |

| Region | BGC province | Biogeochem. model | Physical model | Vert. res. | Horiz. res. | | | | | | | # | # | # | Reference |
|---|---|---|---|---|---|---|---|---|---|---|---|---|---|---|---|
| Arctic | Seasonally stratified | REcoM2 | FESOM | | | + | | | + | | + | 2 | 1 | | (Schourup-Kristensen et al., 2018) |
| Arctic | Seasonally stratified | BLING | NEMO | | | + | | + | | + | | | | | (Castro de la Guardia et al., 2019) |
| Arctic | Seasonally stratified | DARWIN | MITgcm | | | + | + | + | + | | + | 5 | 2 | | (Manizza, 2019) |
| Pacific | Tropical | Leonard | OGCM | | | + | + | | | | | 1 | 1 | 1 | (Christian et al., 2001) |
| Pacific | Tropical | Chai | ROMS | | | | + | | + | | | 2 | 2 | 2 | (Xiu and Chai, 2011) |
| Pacific | Subtropical | Fasham | ROMS | | | | + | + | | | | 1 | 1 | 2 | (Gan et al., 2014) |
| Pacific | Tropical | Fasham | ROMS | | | | + | | | + | + | 1 | 1 | 2 | (Ji et al., 2017) |
| Pacific | Tropical | PISCES | ROMS | | | + | + | + | + | | | 2 | 2 | 3 | (Vergara et al., 2017) |
| Pacific | Tropical | TOPAZ | GOTM | | | | + | + | + | + | + | 3 | | 1 | (Jung et al., 2019b) |
| Pacific | Tropical | Chai | ROMS | | | | + | + | + | + | + | 2 | 2 | 3 | (Ma et al., 2019) |
| Pacific | Tropical | Fasham | ROMS | | | | + | + | | | | 1 | 1 | 2 | (Lu et al., 2020) |
| Pacific | Tropical | Kearney | Bering 10K ROMS | | | + | + | | | | | 2 | 5 | 2 | (Kearney et al., 2020) |

*For the nutrients: dark orange: 0.8<r<1; Bias: < 0.5; RMSD <0.2, Medium orange: 0.5 <r <0.8; 0.5<Bias< 1; 0.5< RMSD <0.2 and Light orange: r <0.5; Bias > 1; RMSD >0.5. For the vertical resolution: dark blue: > 50 layers, medium blue: 20<layers< 50, light blue: < 20 layers.  For the horizontal resolution: dark blue: < 0.1degrees, medium blue: 0.1<degrees<0.5 and light blue: >0.5 degrees.

---

## Author Comment (AC2)

**Reply to reviewers' comments**

Title: "Reviews and syntheses: Assessment of Biogeochemical Models in the Marine Environment"

To the reviewer

> *The authors would like to thank the reviewer for his useful comments that help to improve the manuscript. The authors have considered all suggestions and addressed the raised issues trying to provide necessary clarifications and improvements in the revised manuscript. Based on the reviewer's comments, the authors have started rewriting the three sections (introduction, model approaches, applications of biogeochemical models in the global oceans) as per the reviewers' suggestion and requests.*

> *Below are given point by point answers to the comments (Reviewers comments in normal fonts, response in italic fonts). All the changes will be implemented in the revised manuscript.*

**Comments from the editors and reviewers:**

Reviewer Comments:
Reviewer 2

I do not think that this paper meets the expected standard for a Reviews and Syntheses paper in this journal. The Abstract, the Introduction and the Conclusion mostly fail to explain to the reader what the authors' overall purpose is. The Results are a rather verbose and poorly organized recitation of many details without much of an overall coherent structure. First, one might want to step back and ask why another review on this topic is needed and what differentiates it from existing ones. Ideally this question should be addressed in the first paragraph of the Introduction. This paper never addresses it at all. Secondly, why a review rather than a primary research contribution? There are today a great variety of ocean biogeochemistry data products that are in the public domain and available to anyone with an internet connection. Why not attempt some systematic evaluation of a set of biogeochemistry models against one or more of these, similar to past experiments in the cited literature (e.g., Friedrichs et al, 2007; Kwiatkowski et al, 2014). Instead, this review offers a subjective and meandering assessment of the existing literature and various authors' conclusions regarding how well their models simulate various observables (e.g., in Section 3.1). If the paper were clearly organized around a set of questions or metrics I might be more charitable, but it is not.

> *The authors would like to thank the reviewer for his comments. The authors believe that an updated review on the biogeochemical models is highly needed and would be very useful for the community. Indeed, there is still a critical need to identify the biogeochemical models that are geographically portable and can be applied across diverse ecosystems. Thus, introducing an updated general review on the biogeochemical modeling is also important to review the state of art and help in improving the uncertainties of the existing biogeochemical models.*

> *Therefore, the authors have updated all the sections, especially section 3. They have added more discussions on the applications of the biogeochemical models. The authors have also divided the regions into four zones including: the tropical zone, subtropical zone, seasonally stratified and High Nutrient and Low Chlorophyll (HNLC) zone. The authors have discussed the reasons for the models to perform well in diverse regions and the corresponding physical settings. In the discussion, the authors emphasized that the complex models are not necessarily more accurate*

*than simpler PFTs or NPZD models. For example, both simple NPZD (e.g. Fasham-ROMS) and more complex models (e.g. Darwin and other PFTs) can provide closer estimates for nutrients (e.g. nitrogen) to observations in subtropical and tropical seas as the coastal and continental shelves in these regions are nutrients enriched with organic matter. However, the NPZD-Fasham model is unable to represent the diverse PFTs observed in regions such as the HNLC and the seasonally stratified (Arctic) regions. Therefore, PFTs-based models such as PISCES and TOPAZ coupled to NEMO give better chlorophyll estimates as compared to BLING-NEMO in the seasonally stratified (Arctic) and HNLC (Southern Ocean) zones. Together with PISCES and TOPAZ, PlankTOM also provides a good estimate of chlorophyll in the HNLC zone (Southern Ocean).*

*Section 3 includes extended discussions about the applications of biogeochemical models across diverse ecosystems and Table 1 (attached at the end) is added to support the discussions.*

The classification of models seems very subjective and arbitrary. Why can not a carbon-cycle model incorporate either an NPZD model or a PFT model? Indeed, most modern ones do. The history of the field is the gradual replacement of simple HamOCC or OCMIP type approaches to parameterizing uptake and remineralization of carbon and nutrient (usually P) with explicit biology models. The question is which biology model, and how much complexity is justified and useful? I do not believe that this review sheds much light on this history or offers new and useful information that could help to guide such choices in the future.

*The authors highly appreciate the reviewer's comments. The subsections of the model approaches have been updated to include: 1. Classical NPZD Approach, 2. Adding more elements (other nutrients, and carbon, 3. Increasing the biological complexity and 4. The addition of several classes of organic matter and bacteria as suggested by reviewer 1.*

The only part of this paper that really contains anything new is Table 5. This Table contains a huge amount of information. The only way I can see to salvaging this effort is to reorganize this information around some kind of coherent structure. The nearest historical precedent I can think of is Totterdell (1993, in Evans and Fasham, eds, "Towards a Model of Ocean Biogeochemical Processes") (10.1007/978-3-642-84602-1_15). But the current version seems more like a "data dump". The authors have gleaned a great deal of information from the existing literature. But to justify publication of such a review they need to present it in a way that is useful to the reader.

*The authors highly appreciate the reviewer's comments. They have updated the text of section 3 in which the applications as well as assessment of the models have been discussed. The table has been also updated and it is attached at the end of this document. A figure that shows the statistical performance of the biogeochemical to estimate the Chl-a in the regional oceans has been also added. The table has been also discussed in term of the structure of the biogeochemical models along with the physical settings, elements, living/non-living components, regions and applicability.*

On a personal note, I will tell the authors that early in my career I had a similar paper rejected by a journal editor. It is never a pleasant experience, but we can learn from it.

*The authors thank the reviewer for his advice and comments which help the authors to make substantial changes and improvements in the manuscript.*

[Figure]

The coupled biogeochemical-physical models applied in the regional seas to reproduce the surface chlorophyll-*a* concentrations along with their statistical performance. Red color represents high statistical correlation (r> 0.8), orange color represents medium statistical correlation (0.5 < r < 0.8, and blue represent low statistical correlation (r <0.5).

Table 1. Overview of the biogeochemical models reviewed in this work showing the model approach, physical model, resolution, Nutrient/element cycling and the number of the living/non-living components.

| Ocean | Zone | Model approach | Physical model | Resolution | | Nutrient/element cycling | | | | | | Living/non-living components | | | | Ref. |
|---|---|---|---|---|---|---|---|---|---|---|---|---|---|---|---|---|
| | | | | Vertical layers | Horizontal resolution | Fe | N | P | Si | O$_2$ | C | #of phytoplankton | # of zooplankton | # of detritus | # of Bacteria | |
| Global | | NOBM | GCM | | | + | + | | + | | | 4 | 1 | 2 | | (Gregg et al., 2003) |
| Global | | HAMOCC5 | LSG | | | + | + | + | + | | | 2 | 2 | 1 | | (Aumont et al., 2003) |
| Global | | ERSEM | GOTM | | | + | + | + | + | + | + | 4 | 3 | 1 | 1 | (Blackford et al., 2004) |
| Global | | Moore | CCSM | | | + | + | + | + | + | + | 4 | 1 | 1 | | (Moore et al., 2004) |
| Global | | Moore | CCSM | | | + | + | + | + | + | + | 4 | 1 | 2 | | (Moore & Doney, 2007) |
| Global | | PlankTOM | NEMO | | | + | + | | + | + | + | 5 | 2 | 2 | | (Buitenhuis et al., 2013) |
| Global | | PISCES | NEMO | | | + | + | + | + | + | + | 2 | 2 | | | (Aumont et al., 2015) |
| Global | | DARWIN | MITgcm | | | | + | + | + | + | + | 9 | 2 | 2 | | (Dutkiewicz et al., 2015) |
| Global | | PlankTOM | NEMO | | | | + | | | + | + | 6 | 3 | | 1 | (Andrews et al., 2017) |
| Global | | PISCES | NEMO | | | + | + | + | + | | + | 2 | 2 | 2 | | (Aumont et al., 2017) |
| Global | | Moore | NCAR-CSM1 | | | + | + | + | + | + | + | 4 | 1 | 2 | | (Pant et al., 2018) |
| Global | | TOPAZ/PISCES | NEMO | | | + | + | + | + | + | + | TOPAZ:3 PISCES: 2 | PISCES:2 | | | (Jung et al., 2019a) |
| Atlantic | Tropical | Fasham | MOM | | | | + | | | | | 1 | 1 | 1 | | (Oschlies and Garçon, 1999) |
| Atlantic | Subtropical | Fasham | ROMS | | | | + | | | | + | 1 | 1 | 1 | | (Fennel et al., 2008) |
| Atlantic | Subtropical | Fasham | ROMS | | | | + | | | | + | 1 | 1 | 2 | | (Druon et al., 2010) |
| Atlantic | Subtropical | Fasham | ROMS | | | | + | | | | | 1 | 1 | 2 | | (Xue et al., 2013) |
| Atlantic | Subtropical | ERSEM | OGCM-MED16 | | | + | + | + | + | | | 4 | 4 | 1 | 1 | (Lazzari et al., 2016) |
| Atlantic | Subtropical | ERSEM | POM | | | | + | + | + | + | + | 4 | 3 | 2 | 1 | (Kalaroni et al., 2019) |
| Atlantic | Subtropical | ERSEM | POM | | | + | + | + | + | + | + | 4 | 3 | 2 | 1 | (Kalaroni et al., 2020) |
| Indian | Tropical | Fasham | OGCM | | | + | | | | | | 1 | 1 | 1 | 1 | (Ryabchenko et al., 1998) |

| Region | Zone | Model | Physical model | | | | | | | | | N1 | N2 | N3 | N4 | Reference |
|---|---|---|---|---|---|---|---|---|---|---|---|---|---|---|---|---|
| Indian | Tropical | ERSEM | Princeton/Mellor–Yamada | | | | + | | + | | | 4 | 2 | 2 | 1 | (Blackford and Burkill, 2002) |
| Indian | Tropical | McCreary | Four-layer model | | | | + | | | | | 1 | 1 | 1 | | (Hood et al., 2003) |
| Indian | Tropical | Fasham | MOM | | | | + | | | | | 1 | 1 | 1 | | (Kawamiya and Oschlies, 2003) |
| Indian | Tropical | PISCES | NEMO | | | + | + | + | + | | | 2 | 2 | 3 | | (Koné et al., 2009) |
| Indian | Tropical | PISCES | NEMO | | | | + | | + | | | 2 | 2 | 3 | | (Resplandy et al., 2011) |
| Indian | Tropical | McCreary | Six-layer model | | | | + | | | + | | 1 | 1 | 2 | | (McCreary et al., 2013) |
| Indian | Tropical | Fasham | ROMS | | | | + | | | + | | 1 | 1 | 1 | | (Lachkar et al., 2017) |
| Indian | Tropical | ERSEM | GOTM | | | | + | + | + | + | | 4 | 3 | 1 | 1 | (Sankar et al., 2018) |
| Indian | Tropical | Fasham | ROMS | | | | + | | | + | | 1 | 1 | 1 | | (Lachkar et al., 2019) |
| Indian | Tropical | PISCES | ROMS | | | + | + | + | + | | | 2 | 2 | 3 | | (Guieu et al., 2019) |
| Indian | Tropical | NOBM | OGCM | | | | + | | | | | 4 | 1 | 3 | | (Das et al., 2019) |
| Indian | Tropical | TOPAZ | MOM | | | + | + | + | + | | | 3 | | 2 | | (Sharada et al., 2020) |
| Indian | Tropical | Fasham | ROMS | | | | + | | | + | | 1 | 1 | 1 | | (Lachkar et al., 2020) |
| Southern | HNLC | PlankTOM | NEMO | | | | + | + | + | + | + | 6 | 3 | 3 | 1 | (Le Quéré et al., 2016) |
| Southern | HNLC | NOBM | OGCM | | | | + | | + | | + | 4 | 1 | 3 | | (Trull et al., 2018) |
| Southern | HNLC | DARWIN | MITgcm | | | + | | | | | | 2 | 2 | | | (Uchida et al., 2019) |
| Southern | HNLC | PISCES | NEMO | | | + | + | + | + | | | 2 | 2 | 2 | | (Person et al., 2019) |
| Sothern | HNLC | DARWIN | MITgcm | | | | + | + | + | | + | 6 | 2 | | | (Lo et al., 2019) |
| Southern | HNLC | Chai | ROMS | | | + | + | | + | | | 2 | 2 | 3 | 1 | (Jiang et al., 2019) |
| Southern | HNLC | TOPAZ | ESM2M | | | | + | | | | + | 3 | | 2 | | (Bronselaer et al., 2020) |
| Arctic | Seasonally stratified | PISCES | MITgcm | | | | | | | | + | 2 | 2 | 2 | | (Manizza et al., 2011) |
| Arctic | Seasonally stratified | 21 biogeochemical models [c] | - | | | | | | | | | | | | | (Babin et al., 2016) |

| Ocean | Stratification | Ecosystem model | Physical model | Vert. res. | Horiz. res. | | | | | | | | | | Reference |
|---|---|---|---|---|---|---|---|---|---|---|---|---|---|---|---|
| Arctic | Seasonally stratified | REcoM2 | FESOM | | | + | | | + | | + | 2 | 1 | | (Schourup-Kristensen et al., 2018) |
| Arctic | Seasonally stratified | BLING | NEMO | | | + | | + | | + | | | | | (Castro de la Guardia et al., 2019) |
| Arctic | Seasonally stratified | DARWIN | MITgcm | | | + | + | + | + | | + | 5 | 2 | | (Manizza, 2019) |
| Pacific | Tropical | Leonard | OGCM | | | + | + | | | | | 1 | 1 | 1 | (Christian et al., 2001) |
| Pacific | Tropical | Chai | ROMS | | | | + | | + | | | 2 | 2 | 2 | (Xiu and Chai, 2011) |
| Pacific | Subtropical | Fasham | ROMS | | | | + | + | | | | 1 | 1 | 2 | (Gan et al., 2014) |
| Pacific | Tropical | Fasham | ROMS | | | | + | | | + | + | 1 | 1 | 2 | (Ji et al., 2017) |
| Pacific | Tropical | PISCES | ROMS | | | + | + | + | + | | | 2 | 2 | 3 | (Vergara et al., 2017) |
| Pacific | Tropical | TOPAZ | GOTM | | | | + | + | + | + | + | 3 | | 1 | (Jung et al., 2019b) |
| Pacific | Tropical | Chai | ROMS | | | | + | + | + | + | + | 2 | 2 | 3 | (Ma et al., 2019) |
| Pacific | Tropical | Fasham | ROMS | | | | + | + | | | | 1 | 1 | 2 | (Lu et al., 2020) |
| Pacific | Tropical | Kearney | Bering 10K ROMS | | | + | + | | | | | 2 | 5 | 2 | (Kearney et al., 2020) |

*For the nutrients: dark orange: 0.8<r<1; Bias: < 0.5; RMSD <0.2, Medium orange: 0.5 <r <0.8; 0.5<Bias< 1; 0.5< RMSD <0.2 and Light orange: r <0.5; Bias > 1; RMSD >0.5. For the vertical resolution: dark blue: > 50 layers, medium blue: 20<layers< 50, light blue: < 20 layers. For the horizontal resolution: dark blue: < 0.1degrees, medium blue: 0.1<degrees<0.5 and light blue: >0.5 degrees.